# Explaining Interactions Between Text Spans

**Sagnik Ray Choudhury**[1*]**, Pepa Atanasova**[2*]**, Isabelle Augenstein**[2]
[1]University of Michigan, [2]University of Copenhagen
sagnikrayc@gmail.com, pepa@di.ku.dk, augenstein@di.ku.dk

## Abstract

Reasoning over spans of tokens from different parts of the input is essential for natural language understanding (NLU) tasks such as fact-checking (FC), machine reading comprehension (MRC) or natural language inference (NLI). However, existing highlight-based explanations primarily focus on identifying individual important tokens or interactions only between adjacent tokens or tuples of tokens. Most notably, there is a lack of annotations capturing the human decision-making process w.r.t. the necessary interactions for informed decision-making in such tasks. To bridge this gap, we introduce *SpanEx*, a multi-annotator dataset of human span interaction explanations for two NLU tasks: NLI and FC. We then investigate the decision-making processes of multiple fine-tuned large language models in terms of the employed connections between spans in separate parts of the input and compare them to the human reasoning processes. Finally, we present a novel community detection based unsupervised method to extract such interaction explanations from a model's inner workings.[1]

## 1 Introduction

Large language models (LLMs) employed for natural language understanding (NLU) tasks are inherently opaque. This has necessitated the development of explanation methods to unveil their decision-making processes. Highlight-based explanations (Atanasova et al., 2020; Ding and Koehn, 2021) are common: they produce importance scores for each token of the input, indicating its contribution to the model's prediction. In many NLU problems, however, the correct label depends on interactions between tokens from separate parts of the input. For instance, in fact checking (FC), we verify

---

[*]The first two authors contributed equally.

[1] ⚙ We make the code and the dataset available at: https://github.com/copenlu/spanex
🤗 The dataset is also available at https://huggingface.co/datasets/copenlu/spanex

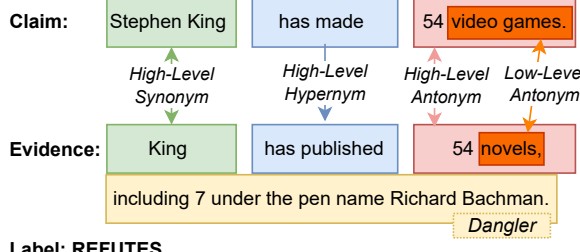

**Claim:** Stephen King | has made | 54 video games.

*High-Level Synonym* | *High-Level Hypernym* | *High-Level Antonym* | *Low-Level Antonym*

**Evidence:** King | has published | 54 novels,

including 7 under the pen name Richard Bachman.
*Dangler*

**Label: REFUTES**

**Figure 1:** Human annotations explaining interactions between text spans on an instance from our *SpanEx* dataset, FEVER part. The presence of antonym interactions (high and low-level) between the corresponding spans in the claim and the evidence leads to the label REFUTES.

whether the evidence supports the claim; in natural language inference (NLI), we verify whether the premise entails the hypothesis. We conjecture that the model explanations for such tasks should capture these token interactions as well.

Rationale extraction methods to capture feature interactions used by a model have been proposed before. However, these primarily capture interactions between neighboring tokens (Sikdar et al., 2021) or between tuples of tokens at arbitrary positions in the input (Dhamdhere et al., 2020; Ye et al., 2021). They do not necessarily consider tokens from distinct parts of the input, such as claim and evidence documents, and the token tuples may not necessarily bear meaning on their own. Currently, there is a lack of explainability techniques that unveil interactions among spans belonging to *different parts* of the input, where the spans are comprised of semantically coherent phrases. More importantly, the development of interaction explanation techniques *has not been accompanied by studies of the human decision-making processes employed for multi-part input tasks*. Before extracting interaction explanations, we would want to ensure that such explanations are indeed valid from a human perspective, i.e., a competent reader could also identify such interactions as an explicit reason

behind their decision-making process. Moreover, the lack of such annotations impedes the comparison between extracted model explanations and human decision-making. We address these research gaps by answering three core research questions.

**RQ1**: *What is the human decision-making process for tasks that involve connecting spans from different parts of the input?* To study this, we collect the dataset *SpanEx* consisting of 7071 instances annotated for span interactions (described in §2; see Fig. 1 for an example annotation). *SpanEx* is the first dataset with human phrase-level interaction explanations with *explicit labels for interaction types*. Moreover, *SpanEx* is annotated by three annotators, which opens new avenues for studies of human explanation agreement – an understudied area in the explainability literature. Our study reveals that while human annotators often agree on span interactions, they also offer complementary reasons for a prediction, collectively providing a more comprehensive set of reasons for a prediction.

**RQ2**: *Do fine-tuned LLMs follow the same decision-making as the human annotators on tasks with multi-part inputs?* *SpanEx* enables an investigation of the alignment between LLMs and human decision-making. We evaluate the sufficiency and comprehensiveness of the human explanations (see §3) for six LLMs. We find that the models rely on interactions that are consistent with the human decision-making process. Interestingly, the models depend more on the interactions where the inter-annotator agreement is high, indicating an inductive bias similar to that observed in humans.

**RQ3**: *Can one generate semantically coherent span interaction explanations?* We propose a novel approach for generating interaction explanations that connect textual spans from different parts of the input (see §4). The generated explanations can contain spans in addition to single tokens, as an explanation consisting of groups of arbitrary tokens would lack meaning for end users (Chen, 2021).

## 2   Span Interaction Dataset

### 2.1   Manual Annotation Task

**Datasets.** We collect explanations of span interactions for NLI on the SNLI dataset (Bowman et al., 2015) and for FC on the FEVER dataset (Thorne et al., 2018). SNLI contains instances consisting of premise-hypothesis pairs, where a model has to predict whether they are in a relationship of entailment (the premise entails the hypothesis), contradiction

(the hypothesis contradicts the premise), or neutral (neither entailment nor contradiction holds). FEVER contains instances consisting of claim-evidence pairs, where one has to predict whether the evidence supports the claim, refutes the claim or there is not enough information (NEI). From here on, we will use *'Entailment'* to denote both 'entailment' and 'supports' labels, *'Contradiction'* to denote both 'contradiction' and 'refutes' labels, and *'Neutral'* to denote both 'neutral' and 'NEI' labels. We will also use 'Part 1' to denote the premise for NLE or the evidence for FC; 'Part 2' to denote the hypothesis for NLI or the claim for FC.

The FC task involves retrieving evidence sentences from Wikipedia articles as the initial step, followed by label prediction. As we are interested only in the interaction between the claim and the evidence parts, we focus only on the second task and use the evidence sentences provided as gold annotations in the original FEVER dataset. For claims with no supporting evidence sentences (NEI class), we employ the well-performing system by Malon (2018) to collect sentences close to the claims.[2]

We collect annotations for a random subset of the test splits of both datasets. While our analysis necessitates the collection of annotations for test instances, we also collect annotations for a random subset of 1100 training instances from each dataset. The latter opens new avenues for studies including span interactions in the training processes of LLMs.

**Interaction Spans.** We introduce the notion of *span interactions*, where the spans are *contiguous parts of the input sufficient to bear meaning*. For an interaction, one span is selected from Part 1 and one from Part 2 of the input. We annotate the spans at both *high and low levels*.

A *high-level span* is the largest contiguous sequence of tokens that i) is *not* the whole part, (i.e., not the entire premise or hypothesis) ii) bears meaning in itself, and iii) can be associated with a span from another part using one of the defined relations. As an example, consider the premise "Two women are running" and the hypothesis "Two men are walking". The label is "contradiction", which has to be justified through the interactions of the constituents in the sentences. We see that the subjects, made out of noun phrases are antonyms: "two men" (premise) and "two women" (hypothesis), and so

---

[2]16.8% of the FEVER instances that contain more than one evidence sentence for a claim are discarded as they require interaction explanation annotations between the evidence sentences themselves. We leave this for future work.

are the predicates, made out of verb phrases: "are running" and "are walking". These are the largest meaningful constituents where such relationships can be established. Therefore, they are considered "high-level" spans. A *low-level span* is the smallest meaning-bearing text span that still holds a relation. For the given example, these would be "man"/"woman", "running"/"walking".

For annotating high-level span boundaries, the annotators were shown the constituency parse tree as a suggestion, but it was not enforced that the boundaries *must* adhere to the constituents, as the semantic segmentation of a sentence does not always adhere to the syntactic one. Annotators first annotated spans at high-level and if smaller spans inside the high-level ones could still hold an interaction with coherent semantics, they proceeded to annotate a low-level span interaction (see an example from the annotation platform in Fig. 6).

**Interactions.** We introduce three types of interactions: 'Synonym', 'Hypernym', and 'Antonym'. A span is a 'synonym' for another one when both denote the same concept, e.g., "two young children" and "two kids". 'Antonym' denotes the opposite, e.g., "one tan girl" and "a boy". 'Hypernym' indicates superordinate interactions, e.g., "a couple" is a hypernym of "two married people" as people can be a couple without getting married but the reverse is not true. While 'Synonym' and 'Antonym' interactions are symmetric, 'Hypernym' interactions have a directional aspect, hence, we use two distinct types: 'Hypernym-P2-P1' and 'Hypernym-P1-P2' depending on whether the hypernym appears within Part 1 or Part 2.

The interaction types defined above are well situated in previous work. The ideal approach to NLI and FC would be to translate Part 1 and Part 2 into formal meaning representations such as first-order logic, but often such full semantic interpretation is unnecessary as pointed by MacCartney and Manning (2014). Consequently, the authors developed a calculus of natural logic based on an inventory of entailment relations between phrases - entailment labels can be inferred based on these relations instead of producing a full semantic parse. Similarly, Yanaka et al. (2019) used the concept of upward and downward entailment.

It can be easily seen that for the Contradiction label, there has to be at least one Antonym interaction: a span must appear in Part 2 that directly contradicts a span in Part 1. This interaction is

the same as the "negation" and "alternation" relations in MacCartney and Manning (2014). For the Entailment label, there should be Synonym interactions ("equivalence" in MacCartney and Manning (2014)), or Part 1 should be more specific. For example, consider an instance with Part 1 as "All workers joined for a French dinner." and Part 2 as "All workers joined for a dinner.". "French dinner" is a true description of "dinner" (but not the other way) because "dinner" is more generic, so Part 1 entails Part 2. In other words, this upward (Yanaka et al., 2019) or forward (MacCartney and Manning, 2014) entailment ("French dinner" → "dinner") should only happen from Part 2 to Part 1 - for our case, a Hypernym-P2-P1 interaction should exist. This also implies that a Hypernym-P1-P2 interaction (downward (Yanaka et al., 2019) or backward (MacCartney and Manning, 2014) entailment) would make the label Neutral. However, one can also create a neutral hypothesis by creating text that has no synonym, hypernym, or antonym relation with a premise span. As described below, these are called Dangler-SYS-P2 interactions ("independence" relations in MacCartney and Manning (2014)) in our setup. In summary, Antonym interactions are important for the Contradiction labels, Hypernym-P2-P1 and Synonyms are important for the Entailment label, and Hypernym-P1-P2 and Dangler-SYS-P2 are important for the Neutral label. The same interactions are used for FEVER as the SNLI labels can be easily mapped to them.

To reduce the annotation load, we asked the annotators not to annotate synonyms (both at low and high levels) where there is a surface-level match (e.g., 'King' appearing in both the claim and the evidence in the example on Fig. 1). We automatically add them to the final version of the dataset (Synonym-SYS interaction).[3] We also add spans that have not been annotated by an annotator as Dangler-SYS-P1 and Dangler-SYS-P2 interactions, depending on their location in Part 1 or Part 2. These are spans that cannot be matched with any span in the other part. They are particularly important if found in Part 2 as they reveal spans that are not supported/entailed by spans in Part 1, leading to the Neutral class.

---

[3]To reduce false positive surface matches, we removed those where both spans include only stopwords. In addition, we took care of false negative surface matches as we found that the original datasets have few instances with morphological errors (e.g., spelling mistakes, and missing apostrophes). We instructed annotators to include those in their annotations.

| Dataset | Entailment | Neutral | Contradiction | Total |
|---|---|---|---|---|
| SNLI | 1298 | 1287 | 1280 | 3865 |
| FEVER | 945 | 1270 | 991 | 3206 |

**Table 1:** Overview of the number of annotated instances from SNLI and FEVER in our *SpanEx* dataset per instance label – "Entailment", "Neutral", "Contradiction".

| Interaction | SNLI | | FEVER | |
|---|---|---|---|---|
| | Low | High | Low | High |
| Synonym | 2719 | 8408 | 1033 | 19785 |
| Hypernym-P1-P2 | 1594 | 2711 | 127 | 572 |
| Hypernym-P2-P1 | 2633 | 6351 | 1337 | 2724 |
| Antonym | 3941 | 4407 | 3325 | 3029 |
| Synonym-SYS | 17872 | 1866 | 37109 | 9939 |
| Dangler-SYS-P1 | 4186 | 10151 | 10922 | 14374 |
| Dangler-SYS-P2 | 7225 | 2542 | 5517 | 4211 |

**Table 2:** Annotated interactions for SNLI and FEVER test splits in *SpanEx*, for low- and high-level spans. Table 6 in App. presents a detailed breakdown by instance label.

**Annotation Task.** Each annotator is provided with an instance from FEVER or SNLI, together with its gold label. The gold label is provided so the annotators can find span interactions in accordance with the label at hand. For example, *Synonym* interactions can be found in instances of all labels. Instances of the Entailment class should have all spans in Part 2 be entailed by spans in Part 1. Hence, all tokens of Part 2 should be part of a *Synonym* or a *Hypernym-P2-P1* interaction with tokens in Part 1. *Antonym* interactions can be annotated only in instances with label Contradiction and at least one Antonym interaction has to be annotated for those. In instances with label Neutral, at least one *Hypernym-P1-P2* interaction or a *dangler in Part 2* has to be found. The above rules are also used for quality control of the annotations where instances that do not contain the necessary or allowed interactions are returned for correction. For detailed annotation guidelines see App. B.

Each instance was annotated by three professional annotators[4] with university education and fluent English language skills, one being a native English speaker. The annotators were trained on 200 instances from each dataset, with two feedback sessions, before annotating the main batch. The annotations were done using the brat tool[5] (see example screenshots in App., Fig.6).

## 2.2 Dataset Analysis

Table 1 shows the instance distribution across labels in *SpanEx*. In total, the dataset consists of 7071 instances, roughly equally distributed between the two tasks and in turn the three labels each. Table 2 gives an overview of the annotated span interactions. Interestingly, we find a higher frequency of annotations for high-level interactions than for low-level ones. This is because spans smaller than the annotated high-level ones are not always semantically coherent. At the high level, the number of Synonym interactions is the highest because they can appear in instances of any label. At the low

[4]https://www.data-bee.net/
[5]https://brat.nlplab.org/

level, the Antonym interaction annotations are the most frequent. We conjecture this is because there are no exact matches or danglers we can annotate automatically for this relation.

Table 3 presents the length of the interaction spans. At high-level, the span length varies from 2.49 to 6.48 tokens on average. For high-level interactions, the Antonym interaction requires the longest spans, while the Synonym interaction – the shortest. We see that the spans usually have a length of one token at the low level.

Finally, Table 4 presents information about the inter-annotator agreement (IAA) in annotating the spans and interactions for *SpanEx*. When considering exact matches of span tuples constituting an interaction annotated by the different annotators, we observe numerous instances where the span tuples annotated by one annotator do not match with those of other annotators (#Span Agree = 1) due to small differences in tokens included. Therefore, we also compute Relaxed Span Match agreement, where we consider span tuples as matching if both the Part 1 span and the span Part 2 have at least one matching token. With the relaxed matching, we find that most spans are annotated by three annotators at the high level and at least by two at the low level. We conjecture that the Relaxed Span Matching is less applicable at the low level, where the annotated span interactions of the different annotators are rather complementary. Finally, the IAA for the span interaction type is significant – up to 91.89 Fleiss' $\kappa$ for low-level FEVER annotations. The IAA for span interaction type resulting from the Relaxed Spans Match remains high, indicating that a large number of the matched interactions are indeed the same spans but with minor token differences. Table 7 in the Appendix presents a more detailed breakdown of the IAA.

| Interaction | SNLI | | FEVER | |
|---|---|---|---|---|
| | **Low** | **High** | **Low** | **High** |
| Synonym | 1.06 | 2.80 | 1.13 | 2.49 |
| Hypernym-P1-P2 | 1.12 | 4.48 | 1.23 | 4.19 |
| Hypernym-P2-P1 | 1.12 | 3.77 | 1.29 | 6.21 |
| Antonym | 1.09 | 4.43 | 1.42 | 6.48 |
| Synonym-SYS | 1.0 | 1.0 | 1.0 | 1.0 |
| Dangler-SYS-P1 | 3.35 | 7.32 | 3.68 | 10.72 |
| Dangler-SYS-P2 | 3.85 | 9.48 | 4.56 | 19.20 |

**Table 3:** Span token length per interaction type for SNLI and FEVER test splits in *SpanEx* for low and high-level spans.

| # Anno-tators | Span Level | Exact Spans Match | | Relaxed Spans Match | |
|---|---|---|---|---|---|
| | | # Inter-actions | Interaction type Fleiss' $\kappa$ | # Inter-ations | Interaction type Fleiss' $\kappa$ |
| | | **SNLI** | | | |
| 1 | Low | 4046 | - | 3073 | - |
| | High | 6889 | - | 989 | - |
| 2 | Low | 2560 | - | 2427 | - |
| | High | 3050 | - | 2065 | - |
| 3 | Low | 831 | 86.45 | 1334 | 68.96 |
| | High | 3225 | 84.56 | 5617 | 72.22 |
| | | **FEVER** | | | |
| 1 | Low | 2396 | - | 2099 | - |
| | High | 6715 | - | 1788 | - |
| 2 | Low | 972 | - | 1064 | - |
| | High | 2743 | - | 1584 | - |
| 3 | Low | 589 | 91.89 | 1581 | 87.52 |
| | High | 4730 | 70.93 | 7142 | 74.65 |

**Table 4:** Annotator agreement for *SpanEx*. '# Annotators' indicates the number of annotators that have annotated the interaction. We either do an *exact or relaxed match* of the spans (see §2.2). '# Interactions' indicates the number of interactions that have been annotated by each corresponding number of annotators. 'Interaction type Fleiss $\kappa$' indicates the IAA for interactions annotated by all three annotators.

# 3 Model and Human Explanations Agreement

In §2, we discussed how the annotators modeled the spans and their interactions that led to the classification decision. We next investigate if fine-tuned LLMs use the same decision-making process by comparing the human annotations with two baselines. The **Random Phrase baseline** randomly samples both spans of the interaction from each of the two parts of the input. The **Part Phrase baseline** selects one span from the human annotations, and samples the other one at random from the remaining part. If the models follow the same reasoning as the annotators, the human explanations will have a significantly higher score than the baselines. However, if the annotated interactions are not important for the model, or only one part of the input is sufficient, we will see no such difference.

## 3.1 Evaluation Protocol

Following Chen et al. (2021), we use Area Over the Perturbation Curve (**AOPC**) and Post-hoc Accuracy (**PHA**) to evaluate how faithful model and human explanations are to a model's inner-workings. **AOPC** and **PHA** measure the utility of an explanation $e_i$ for instance $x_i$ by first *removing/adding* the most important $k$ spans from $x_i$ as per $e_i$. This results in a perturbed instance $x_i^{r,k}$ / $x_i^{a,k}$ :

$$x_i^{r,k} = \{x_{i,j} \notin top(e_i, k)\} \tag{1}$$

$$x_i^{a,k} = \{x_{i,j} \in top(e_i, k)\} \tag{2}$$

where $top(e_i, k)$ is a function selecting a set of the top $k$ most important spans according to $e_i$.

**Area Over the Perturbation Curve.** Following instance perturbation, **AOPC** measures the utility of the explanation as the difference between the probability for the originally predicted class $y_i$ given the original instance $x_i$ and the probability for the originally predicted class $y_i$ given the perturbed instance $x_i^{r,k}$ / $x_i^{a,k}$:

$$r(x_i, e_i, k) = p(\hat{y}_i | x_i) - p(\hat{y}_i | x_i^{r,k}) \tag{3}$$

$$a(x_i, e_i, k) = p(\hat{y}_i | x_i) - p(\hat{y}_i | x_i^{a,k}) \tag{4}$$

The function $r$ estimates the effect of *removing $k$ most important spans* from $x_i$. Intuitively, it measures the **comprehensiveness** (**AOPC-Comp**) of the top-k most important spans. If the list of most important spans is comprehensive, it should significantly decrease the predicted probability for the originally predicted class $\hat{y}_i$ when removed. Alternatively, the function $a$ estimates the effect of *preserving only the $k$ most important spans* in the instance $x_i$. It measures the **sufficiency** (**AOPC-Suff**) of the most important $k$ spans in preserving the probability for the originally predicted class $\hat{y}_i$. Finally, **AOPC** measures the overall utility of the explanation by iteratively increasing the number $k \in [0, K]$ of occluded or inserted spans. The results for the separate $k$ values are summarised by a single measure that estimates the area over the curve defined by the results for each $< k, r/a(x_i, e_i, k) >$ pair.

**Post-hoc Accuracy.** Post-hoc accuracy selects one or multiple values for $k$, and computes the *preserved accuracy of a model* for the perturbed instances in the dataset $- X' = \{x_i^{a,k}\}$. This results in a $top - k$-accuracy score (or curve in the case of multiple $k$ values) for one explanation method.

**Adaptation for Span Interactions.** A *better* explanation will have a *higher* **AOPC-Comp** and

**PHA** score and a *lower* **AOPC-Suff** score. However, the annotations do not have an importance score for each span interaction. Hence, the span pairs cannot be ranked, and, in turn, a top-$k$ estimation is not possible. This naturally creates longer explanations, i.e., a higher number of tokens are perturbed. As **AOPC** and **PHA** scores are positively correlated with the number of changed tokens, the baselines can not be fairly compared with the human explanations. Therefore, we normalize the scores by the number of perturbed tokens. Moreover, in the baseline explanations, both the number of span pairs and the number of tokens in each span are sampled from the same distributions as in the annotations.

### 3.2 Experiments, Results & Discussions

Both NLI and FC are multi-class classification tasks. We use linear classifiers on top of pre-trained LLM encoders and fine-tune the models. Six models of the BERT family (Devlin et al., 2019; Liu et al., 2019) varying in model size, tokenization, and pre-training objectives are used: $\text{BERT}_{\text{base-cased}}$, $\text{BERT}_{\text{base-uncased}}$, $\text{BERT}_{\text{large-cased}}$, $\text{BERT}_{\text{large-uncased}}$, $\text{RoBERTa}_{\text{base}}$, and $\text{RoBERTa}_{\text{large}}$. For each model type, we train three models with standard training configurations but a different number of epochs and minimize the Cross-Entropy loss. As the models show little difference in the test data, we choose the best-performing model from each type for the subsequent experiments.

**AOPC-Comp results** (Fig. 2). We merge the Synonym-System and Synonym categories as the annotators would have labeled them the same. In general, high-level interactions have lower scores than low-level ones. This is expected as they are more semantically coherent but they may contain extraneous tokens. The models find the *relevant* low-level annotated interactions, i.e., the ones correlated with human reasoning, more important than: the Random Phrase baseline in all cases; the Part Phrase baseline in all but one case (SNLI-Entailment); and non-relevant interactions in 66% ($\frac{4}{6}$) of the cases. Humans would, e.g., find the Antonym interactions most important for the Contradiction instances, and so do the models. Similarly, for the Neutral ones, the most important interaction found by the models is Hypernym-P1-P2. Moreover, for SNLI, the Dangler-SYS-P2 interactions in the Neutral instances are more

important than the baselines too. An exception is the Entailment class, where we would expect both Hypernym-P2-P1 and Synonym interactions to have higher scores than the baselines and the other interactions, but the Part Phrase baseline has a higher score for both of them. The Hypernym interactions show a large variance as we average over both models and annotators. For the high-level interactions, a similar trend can be observed for the Contradiction and Neutral instances in SNLI. However, for FEVER, we do not observe this; in fact, the baseline scores are mostly higher than the relevant interactions. We hypothesize that the high-level span annotations in the FEVER instances have significantly more extraneous information and possibly can be heuristically shortened in future work.[6]

**Evaluation summary of all metrics.** Explainability evaluation metrics often disagree with each other (Atanasova et al., 2020). Therefore, in Table 5, we summarize (see App. §C for details) how different metrics vary in terms of ranking the relevant interactions. Ideally, the most relevant interactions should be ranked the *highest* by the **AOPC-Comp** and **PHA** scores, and the *lowest* by the **AOPC-Suff** scores. For example, for the Entailment instances in SNLI, the low-level Synonym or Hypernym-P2-P1 interactions are found the most important by **PHA** (indicated by green). None of these interactions is the most important according to the **AOPC-Comp** metric, but it finds at least one of them to be the second most important ( yellow ). **AOPC-Suff**, on the other hand, finds them to be the *least* important ( green ) as expected. In summary, in 64% cases, the relevant interactions are found to be the most (by **AOPC-Comp** or **PHA**, or least, by **AOPC-Suff**) important, in 31% cases they are in the upper (lower) $50^{\text{th}}$ percentile of all interactions, and in 5% cases they are not found relevant. **AOPC-Comp** and **AOPC-Suff** provide complementary evaluations, and they both align well: they differ strongly (indicated by red vs green in the same rows in Table 5) in 4% cases, and moderately in 13% cases (yellow vs green).

**Do all models follow the human decision-making process?** We analyze this in Fig. 3 by comparing the **AOPC-Comp** scores for the three most relevant low-level interactions for three classes: Antonyms for Contradiction, Synonyms

---

[6] In Figure 2, the Part-Phrase baseline uses the random tokens from Part 2. App. figures 9, 10, and 11 show the results when the random tokens are chosen from Part 1. The trends we observe there are similar.

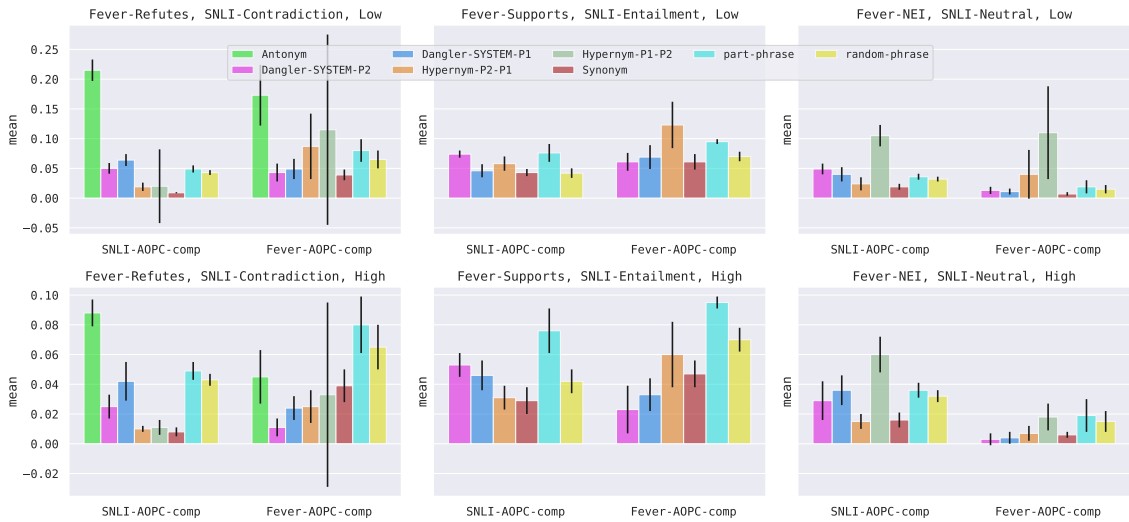

**Figure 2: AOPC-Comp** scores averaged (error bars: standard deviation) over annotators and models. A higher **AOPC** value for relevant interaction (e.g., Antonym for Contradiction) indicates better human-model explanation alignment.

| Dataset | Level | AOPC-Comp | AOPC-Suff | PHA |
|---|---|---|---|---|
| SNLI-Contradict | Low | 1/6 | 6/6 | 3/6 |
| | High | 1/8 | 8/8 | 2/8 |
| SNLI-Entailment | Low | 2/4 | 4/4 | 1/4 |
| | High | 5/6 | 6/6 | 1/6 |
| SNLI-Neutral | Low | 1/5 | 5/5 | 1/5 |
| | High | 1/7 | 7/7 | 1/7 |
| FEVER-Refutes | Low | 1/6 | 5/6 | 5/6 |
| | High | 3/8 | 5/8 | 3/8 |
| FEVER-Supports | Low | 1/4 | 4/4 | 1/4 |
| | High | 3/6 | 6/6 | 2/6 |
| FEVER-NEI | Low | 1/5 | 5/5 | 1/5 |
| | High | 2/7 | 1/7 | 1/7 |

**Table 5:** The rank of the most relevant interactions according to different metrics. The colors green, yellow, and red indicate whether one of the most relevant interactions for a label is the first (**AOPC-Comp** and **PHA**, last for **AOPC-Suff**), in the top 50%, or the bottom 50% (reverse for **AOPC-Suff**) of all interactions.

for Entailment, and Hypernym-P1-P2 interactions for Neutral. For SNLI, we do not see a significant difference, but the BERT$_{base-cased}$ and BERT$_{large-cased}$ models pay the least attention to the relevant interactions in the FEVER Refute and NEI instances. These two models have good F1-scores on the entire test dataset (86.2% and 87.2%, respectively) but very low scores on our annotation instances – 68.9% and 46.8% – whereas all others have > 83% (except 81.1% for RoBERTa$_{base}$, which again has a poor **AOPC-Comp** score for the NEI instances). This further shows that our

method of modeling the human decision process has a strong correlation with the models' reasoning. Similarly, we investigate whether the models depend more on the interactions where the annotators agree. As before, we compute **AOPC-Comp** on the most relevant interactions but split them into: interactions where a) all three annotators agree, b) two annotators agree and c) all disagree. Fig. 4 shows that IAA has a strong correlation with the **AOPC** scores, indicating again that the models have the same inductive biases as humans.

## 4 Extracting Interactive Explanations

An explanation method should output interaction pairs of sets of tokens from the parts of the input $\mathbf{p}^E = \{\langle\{x_i^{part1}\}, \{x_j^{part2}\}, v\rangle | i \in [1, |part1|], j \in [1, |part2|]\}$, where each pair is further assigned a significance value $v$ depending on its influence on the prediction of the model.

Interactions between features, e.g., tokens, in ML models are most commonly learned with an attention mechanism (Vaswani et al., 2017). Hence, we generate a directed bipartite interaction graph $G^I = (V, E)$ with tokens from two parts of the input. The weights for the edges come from the attention matrix. We keep only the attention weights between tokens that belong to different parts of the input, thus, creating two vertex partitions. While we create the interaction graph using attention weights, it can be built using other explanation techniques producing token interaction scores.

We use the top layer attention scores as they dictate how the final representation before the clas-

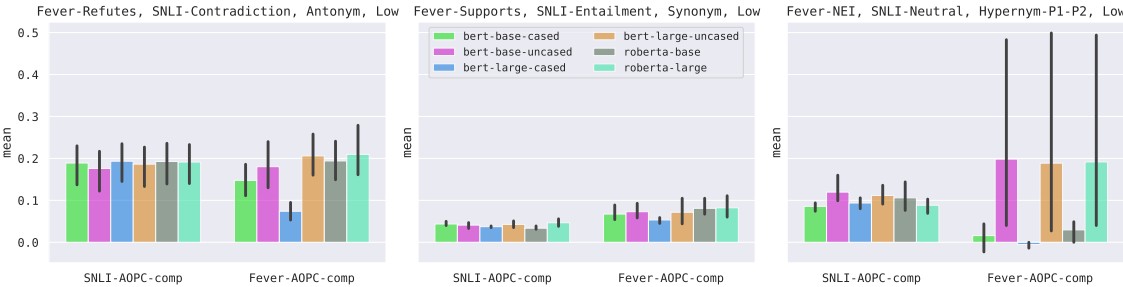

Figure 3: **AOPC-Comp** scores for different models in the most relevant interactions.

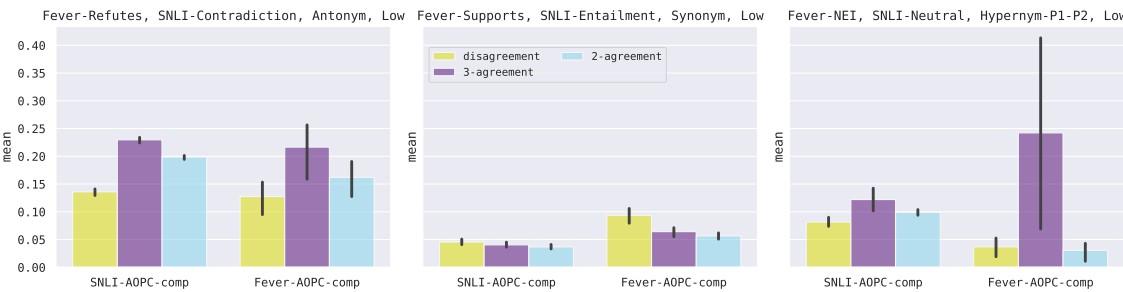

Figure 4: **AOPC-Comp** scores for the most relevant interactions split by inter-annotator agreement.

sification layer is generated. This still leaves us with three choices: a) aggregating the attention matrices from different heads; b) producing a multi-dimensional interaction graph (Tang et al., 2010) where each head will present a different type of interaction; and c) finding the most important head for the classification task. Attention heads should not be aggregated as they are designed to provide different views of the data and capture different semantic and syntactic relations (Rogers et al., 2020). Therefore, we choose to find the most important head using the two methods described next, and we leave the multi-graph approach for future work.

**Classifier Weight.** All our models use a linear classifier ($\mathbf{W}, (n \times m)$, $m$ is the number of classes on top of an n-dimensional `CLS` vector denoted as $\mathbf{c}$.[7] For an input instance $x$ with predicted class index $k$, the logit score for $k$ is a dot product of $\mathbf{w}$ ($W_k^T$) and $\mathbf{c}$, i.e, $s_k = \sum_{i=1}^n w_i c_i$. For all $c_i > 0$, the higher is $w_i$, the higher is $s_k$, and conversely, for all $c_i < 0$, a higher value of $w_i$ makes a higher negative contribution. In summary, the logit score is proportional to $\sum_{i=1}^n sign(c_i).w_i$. We can write the `CLS` vector as $[\mathbf{c_1} \oplus \mathbf{c_2}.. \oplus \mathbf{c_a}]$ and $\mathbf{w}$ as $[\mathbf{w_1} \oplus \mathbf{w_2}.. \oplus \mathbf{w_a}]$ where $a$ is the number of attention heads and $\oplus$ denotes concatenation. Then $s_k$ can be written as $\sum_{j=1}^a \mathbf{w_j}\mathbf{c_j}$. The $\mathbf{w_j}$ with the

highest $\sum_{i=1}^l sign(c_{j,i}).w_{j,i}$ ($l = n//a$, i.e., the dimension of each attention head) makes the highest contribution towards the classification and $j$ is chosen as the most important attention head.

In another approach, **Scalar Mix**, we freeze the parameters of the encoders and train new models with a set of parameters $[\lambda_1..\lambda_a]$ on top of the frozen `CLS` representations. The resulting linear classifier ($\mathbf{W}', (l \times m)$) in this model uses the *scalar mixed* (Peters et al., 2018) `CLS` vector $\sum_{i=1}^a \lambda_i \mathbf{c_i}$. $argmax_i(\lambda_i)$ determines the most important attention head.

We use **community structure detection** algorithms (Fang et al., 2020) on $G^I$ to *find groups of nodes (tokens) with dense inter-group and sparse intra-group connections*. These algorithms are computationally optimized for large social (Gu et al., 2019) and biological networks (Yanrui et al., 2015) and hence overcome the limitation of existing perturbation and simplification-based explanations that rely on the occlusion of groups of input tokens, which leads to a combinatorial explosion when considering span interactions. We use the Louvain algorithm (Blondel et al. (2008), see App. §D) which has been used in directed graphs such as ours. The bipartite nature of $G^I$ ensures that the explanation tokens are from two parts of the input, which are then combined to produce spans based on their positions. Finally, a list of span pairs is

---

[7]We use **boldface upper case** for matrices and **boldface lower case** for vectors.

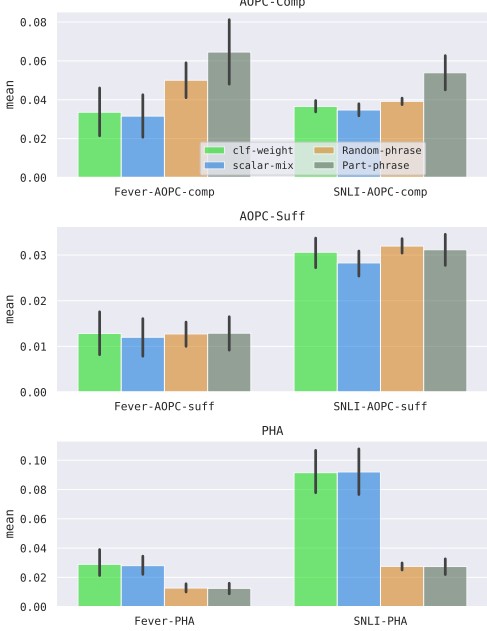

**Figure 5:** Top-3 evaluation scores for Louvain community detection over two types of attention graphs, along with the Part Phrase and Random Phrase baselines. **AOPC-Comp** & **PHA**: higher is better, **AOPC-Suff**: lower is better.

generated by a cartesian product of the generated spans. The score for each span pair is the sum of the edge weights between the nodes in them. The ranked list of span pairs constitutes the explanation.

**Results.** We evaluate the explanations with the same metrics as before (see §3.1) but use the 'top-k' versions. Fig. 5 shows the **top-3 AOPC-Comp**, **AOPC-Suff**, and **PHA** scores for the proposed methods and the baselines (App. §E, Table 10 shows top-1 and top-5 results and some generated explanations). The **PHA** scores are significantly higher in the proposed methods vs. the baselines, and the **AOPC-Suff** scores are lower (as expected) but not greatly so. The baselines do much better in terms of **AOPC-Comp**, which means that the explanations produced by our methods are often sufficient, but not always comprehensive.

## 5 Related Work

Existing work mainly explores interactions between tuples of tokens. Tsang et al. (2020) propose a method to detect grouped pairwise token interactions, where the only interactions occur between tokens in the same group. Hao et al. (2021) creates attention attribution scores using integrated gradient (IG) over attention matrices and then constructs self-attention attribution *trees*. The latter is extended by Ye et al. (2021) to multiple layers.

Several approaches also extend highlight-based explanations to detect interactions between tuples of tokens. One line of work (Tsai et al., 2023; Sundararajan et al., 2020; Grabisch and Roubens, 1999), introduces axioms and methods to obtain *interaction Shapley scores*. Janizek et al. (2021) extend IG by assuming that the IG value for a differentiable model is itself a differentiable function, thus, can be applied to itself. Masoomi et al. (2022) extend univariate explanations to produce bivariate Shapley value explanations. Additionally, Chen et al. (2021) find groups of correlated tokens from different input parts, but the tokens are found at arbitrary positions and the produced explanations are not necessarily semantically coherent. In contrast, we investigate interactions between token spans that bear sufficient meaning and are semantically coherent and thus plausible to end users.

Finally, there is a stream of work on explainability methods for constructing *hierarchical* interaction explanations. Sikdar et al. (2021) compute importance scores in a bottom-up manner starting from the individual embedding dimensions, working its way up to tokens, words, phrases, and finally the sentence. Zhang et al. (2021) build interpretable interaction trees, where the interaction is again defined based on Shapley values. While these methods produce spans of tokens that are part of an interaction, the hierarchical nature of the explanation limits the interactions only to neighboring spans. In contrast, we are interested in spans that can appear in the different parts of the input for NLU tasks and are not necessarily neighboring.

## 6 Conclusion

We introduce *SpanEx*, a multi-annotator explanation dataset that captures the interactions between semantically coherent spans from different inputs in pairwise NLU tasks, here, NLI and FC. *SpanEx* maps the implicit human decision-making process for these tasks to explicit lexical units and their interactions, opening up new research directions in explainability. Using this dataset, we show that fine-tuned LLMs share the human inductive bias as evidenced by their relatively higher scores on established explainability metrics compared to random baselines. We also propose novel community detection-based methods to extract such explanations with modest success. We hope this work will pave the way for further research in the nascent area of interaction explanations.

## Limitations

In this work, we study explanations of span interactions for explaining decisions for NLU tasks. To accomplish this, we have introduced a dataset of human annotations of span interactions for two existing datasets for NLU tasks – fact checking and natural language inference. It is worth noting that there are other NLU tasks such as question answering that necessitate reasoning involving interactions among multiple parts of the input. These tasks may involve different types of interactions, which could be investigated in future work. Furthermore, our dataset consists of interactions between spans from two separate parts of the input. Interactions of more than two spans and from more than two parts of the input are also possible for example in fact checking where interactions between several evidence sentences are possible as well.

In our model analysis, we studied the most popular bidirectional Transformer models. With our dataset and the performed analysis, we have set the ground and only scratched the surface of the prospects to inspect the inner workings of a multitude of different architectures for span interactions, such as auto-regressive Transformer models. The implementation can be easily adapted to perform the following studies in future work.

We have introduced an unsupervised community detection approach for explaining interactions between spans of text, which serves as a foundational step for future research in interactive explanations. However, it is crucial to address the limitations of these initial advancements. Firstly, the explanations produced by the community detection method may consist of spans that lack semantic coherence, as the start or end of a span might not align precisely with the tokens of an exact phrase. Ensuring better semantic coherence within the generated spans is an important aspect to consider for further improvement. Secondly, the current approach does not provide explanations at both the high and low levels, in accordance with the human annotations. Expanding the approach to incorporate explanations at both levels would enhance its completeness and alignment with human annotations. Finally, the method does not explicitly indicate the type of span interaction, such as Antonym, Synonym, or Hypernym. Incorporating the identification of span interaction types would provide valuable information and improve the interpretability of the generated explanations.

## Ethics Statement

The primary objective of our work is to offer span interactive explanations for NLU tasks. The explanations provided by our unsupervised community detection method can be utilized by both machine learning practitioners and non-expert users. It is important to acknowledge the potential risks associated with overreliance on our span interactive explanations as the sole explanation method. Other explanation types, such as free-text explanations (Camburu et al., 2018; Wang et al., 2020; Rajani et al., 2019) can offer complementary information, but their faithfulness could be hard to estimate (Atanasova et al., 2023). Despite these limitations, we believe that our work is an important stepping stone in the area of interactive explanation generation.

## Acknowledgements

This research was co-funded by the European Union (ERC, ExplainYourself, 101077481), by the Pioneer Centre for AI, DNRF grant number P1, as well as by The Villum Synergy Programme. Views and opinions expressed are however those of the author(s) only and do not necessarily reflect those of the European Union or the European Research Council. Neither the European Union nor the granting authority can be held responsible for them. We thank the anonymous reviewers for their helpful suggestions.

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

| Interaction | Entail | Neutral | Contradict | Total |
|---|---|---|---|---|
| **SNLI Low-Level** | | | | |
| Synonym | 1537 | 735 | 447 | 2719 |
| Hypernym-h-to-p | 1770 | 446 | 417 | 2633 |
| Hypernym-p-to-h | 0 | 1529 | 65 | 1594 |
| Antonym | 0 | 0 | 3941 | 3941 |
| Synonym-SYSTEM | 7807 | 5773 | 4292 | 17872 |
| Dangler-SYSTEM-HYPOTHESIS | 1913 | 2375 | 2937 | 7225 |
| Dangler-SYSTEM-PREMISE | 4186 | 2705 | 4511 | 11402 |
| **SNLI High-Level** | | | | |
| Synonym | 3964 | 2490 | 1954 | 8408 |
| Hypernym-h-to-p | 3887 | 1392 | 1072 | 6351 |
| Hypernym-p-to-h | 0 | 2526 | 185 | 2711 |
| Antonym | 0 | 0 | 4407 | 4407 |
| Synonym-SYSTEM | 404 | 1004 | 458 | 1866 |
| Dangler-SYSTEM-HYPOTHESIS | 81 | 1903 | 558 | 2542 |
| Dangler-SYSTEM-PREMISE | 3234 | 3733 | 3184 | 10151 |
| **FEVER Low-Level** | | | | |
| Synonym | 651 | 141 | 241 | 1033 |
| Hypernym-p-to-h | 0 | 117 | 10 | 127 |
| Hypernym-h-to-p | 1123 | 112 | 102 | 1337 |
| Antonym | 0 | 0 | 3323 | 3325 |
| Synonym-SYSTEM | 13549 | 11482 | 12078 | 37109 |
| Dangler-SYSTEM-Claim | 2020 | 1118 | 2379 | 5517 |
| Dangler-SYSTEM-Evidence | 4490 | 1811 | 4621 | 10922 |
| **FEVER High-Level** | | | | |
| Synonym | 6680 | 7081 | 6024 | 19785 |
| Hypernym-p-to-h | 0 | 554 | 18 | 572 |
| Hypernym-h-to-p | 2259 | 339 | 126 | 2724 |
| Antonym | 0 | 0 | 3027 | 3029 |
| Synonym-SYSTEM | 2130 | 5882 | 1927 | 9939 |
| Dangler-SYSTEM-Claim | 9 | 4082 | 120 | 4211 |
| Dangler-SYSTEM-Evidence | 3942 | 6418 | 4014 | 14374 |

**Table 6:** Overview of the number of annotated interactions in our *SpanEx* dataset per instance label.

# A   Detailed Overview of *SpanEx*

Table 6 presents a detailed overview of the annotated interactions. Table 7 presents a detailed overview of the annotated spans.

# B   Annotation Guidelines

Fig. 6 presents a screenshot from the annotation platform with three example annotations.

## B.1   General Description of the NLI Task

You will be provided with <label | premise | hypothesis >, where premise and hypothesis are sentences, which can have one of the three possible labels: entailment, neutral, and contradiction, depending on whether the hypothesis entails the premise. The premise is a caption of an image. The hypothesis was written given the premise, but not the image.

1. Entailment: the hypothesis is definitely a true description of the image: "Two dogs are running through a field.", "There are animals outdoors."

2. Neutral: the hypothesis might be a true description of the image: "Two dogs are running through a field", "Some puppies are running to catch a stick." – the dogs are not necessarily puppies.

3. Contradiction: the hypothesis is definitely a false description of the image: "Two dogs are running through a field.", "The pets are sitting on a couch." – it's impossible for the dogs to be both running and sitting.

## B.2   General Description of the Fact Checking Task

You will be provided with <label | evidence | claim >. The evidence comes from Wikipedia pages and the title of the page is prepended to the sentence (e.g. [source: Islamabad]). The pair can have one of the three possible labels: supports, refutes, not enough info.

## B.3   Overall Description of the Labeling Task

You will be provided with 1) the premise/evidence and the hypothesis/claim and 2) the label for the pair. You will have to find corresponding spans in the premise and the hypothesis and annotate the interaction between them. Our goal is to see how humans determine NLI or FC labels using the interactions between the parts of the premise and hypothesis. There can be different types of interactions, which we define further down below. You will have to find these parts (spans) and label these interaction types.[8]

## B.4   Interaction types

Two corresponding spans – $\alpha \in$ premise and $\beta \in$ hypothesis can have one of the following interactions:

1. Synonym – $\alpha$ denotes the same as $\beta$. Example: pretty and attractive.

2. Antonym – $\alpha$ denotes the opposite of $\beta$. Example: dead and alive; parent and child.

3. Hypernym – $\alpha$ is superordinate to $\beta$ In other words, $\beta$ is more specific than $\alpha$ which can also be due to new details introduced in $\alpha$. Example: an animal is a hypernym of mammal; red is a hypernym of scarlet; to cut is a hypernym of to trim and to slice; 'wash their hands' is a hypernym of 'wash their hands in a sink'.

4. Hypernym-h-to-p – $\alpha$ is in the hypothesis, $\beta$ is in the premise

---

[8]We use 'premise' to denote both premise (NLI) and evidence (FC), we use 'hypothesis' to denote both hypothesis (NLI) and claim (FC).

| Dataset | Num. Ann. | Synonym | Antonym | Hypernym-p-to-h | Hypernym-h-to-p | Total |
|---|---|---|---|---|---|---|
| SNLI-Low | 1 | 661 | 1413 | 919 | 1053 | 4046 |
| SNLI-High | 1 | 1614 | 1953 | 1429 | 1893 | 6889 |
| SNLI-Low | 2 | 640 | 808 | 369 | 743 | 2560 |
| SNLI-High | 2 | 1035 | 631 | 425 | 959 | 3050 |
| SNLI-Low | 3 | 354 | 329 | 23 | 125 | 831 |
| SNLI-High | 3 | 1648 | 420 | 179 | 978 | 3225 |
| SNLI-Low | 1+2+3 | 1655 | 2550 | 1311 | 1921 | 7447 |
| SNLI-High | 1+2+3 | 4297 | 3004 | 2033 | 3830 | 13164 |
| SNLI-Low+High | 1+2+3 | 5952 | 5554 | 3344 | 5751 | 20601 |
| FEVER-Low | 1 | 397 | 1181 | 100 | 718 | 2396 |
| FEVER-High | 1 | 2991 | 1622 | 443 | 1659 | 6715 |
| FEVER-Low | 2 | 232 | 483 | 15 | 242 | 972 |
| FEVER-High | 2 | 1898 | 414 | 63 | 368 | 2743 |
| FEVER-Low | 3 | 102 | 406 | 2 | 79 | 589 |
| FEVER-High | 3 | 4412 | 195 | 5 | 118 | 4730 |
| FEVER-Low | 1+2+3 | 731 | 2070 | 117 | 1039 | 3957 |
| FEVER-High | 1+2+3 | 9301 | 2231 | 511 | 2145 | 14188 |
| FEVER-Low+High | 1+2+3 | 10032 | 4301 | 628 | 3184 | 18145 |

**Table 7:** Number of annotations by type (Synonym, Antonym, Hypernym-P1-P2, Hypernym-P2-P1) and by the number of annotators (1, 2, 3, 1+2+3) that have annotated it.

5. Hypernym-p-to-h – $\alpha$ is in the premise, $\beta$ is in the hypothesis

Note: there can be spans in either part of the instance that do not have a corresponding span in the other part, for short danglers. You can leave these without annotations.

Note: there can be spans with danglers at low-level, with a danglers contained both in premise and hypothesis. In this case, at high-level annotate the two corresponding spans as both Hypernym-h-to-p and Hypernym-p-to-h.

### B.5 How labels define which interactions can be used

Entailment/supports: (mainly synonyms, but premise can be more specific)

- Have synonym interactions;
- Can have danglers (additional information) in the premise;
- Can have Hypernym-h-to-p interactions (more-specific premise).

Neutral/not enough info: (hypothesis has more info/is more specific)

- Have at least one Hypernym-p-to-h OR at least one dangler (when there are only synonym interactions between the premise and the hypothesis) in the hypothesis;

- Can have synonyms, danglers in the premise, Hypernym-h-to-p.

Contradiction/refutes:

- Have at least one antonym interaction;
- Can have hypernym, synonym, dangler interactions;

### B.6 High-Level Text Spans

Annotate interactions between high-level matching text spans in the premise and the hypothesis. Choose a text span $\alpha$ in the premise that can be mapped to a text span $\beta$ in the hypothesis by one of the interactions: synonym, antonym, hypernym. If there's no corresponding high-level span, mark them as Dangler. The spans should be selected at the highest level of the syntax tree, i.e. longest possible chunks that can hold one such interaction. Annotate the two text spans $\alpha$ and $\beta$ as premise and hypothesis. Connect the chunks in the interaction with an arrow. For antonyms and synonyms, which are symmetric interactions, you can draw the interaction arrow starting from the hypothesis. For hypernyms, if the hypernym is located in the premise, start drawing the arrow from the premise, otherwise – from the hypothesis. Annotate the type of the interaction.

Note: If the high-level surface forms match, they still need to be annotated. At high-level span annotation, do not annotate only the dangling parts

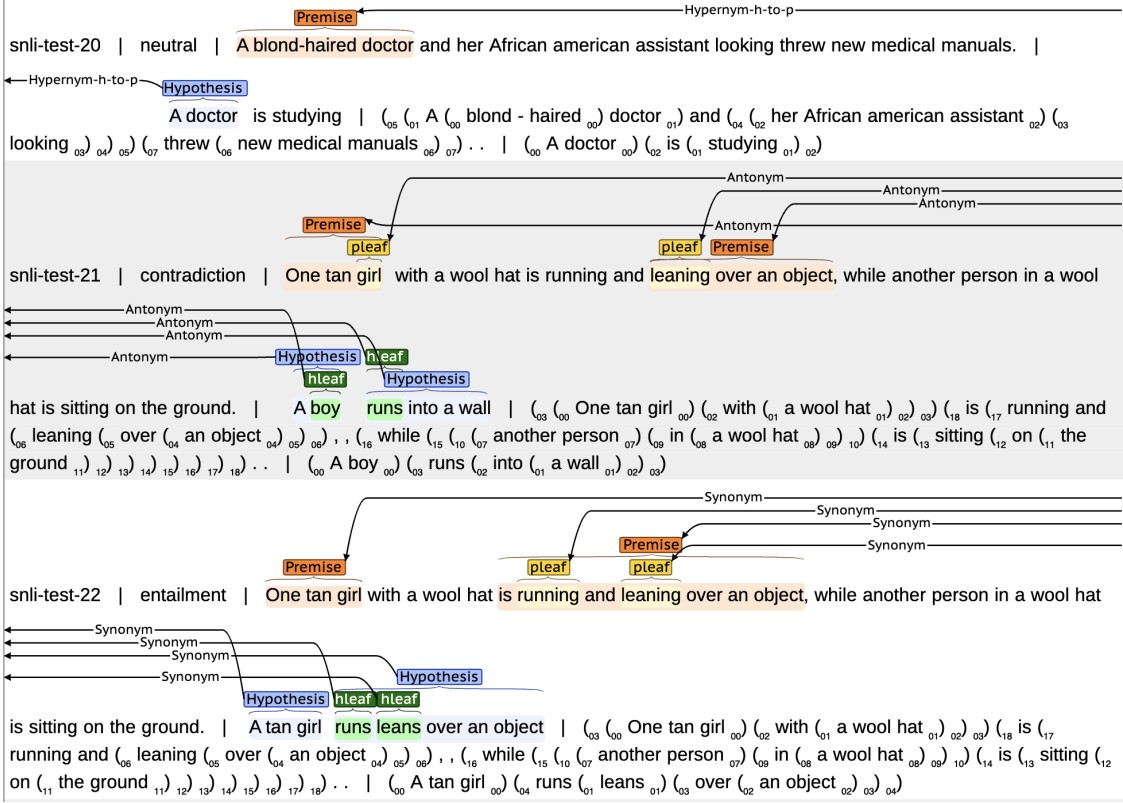

**Figure 6:** Screenshot of the annotation tool.

(those that do not have a corresponding span in the other text part).

Note FEVER: For the FEVER dataset, annotate interactions to the evidence document's title prepended to the evidence in square brackets []. Treat the evidence title and the evidence text as one textual part, i.e. there should be no interactions between the evidence title and the evidence text, but only between the evidence title and the claim and the evidence text and the claim. In such cases, for the same named entity in the claim there will be interactions both to the evidence and to the evidence title.

Note: Annotate all possible interactions of one span to spans in the other part of the text. For example, if one entity can have a corresponding span in the evidence and in the title, annotate both. These include pronouns as well, e.g. Example 9 in Examples FEVER below – 'Sabbir Khan' in the claim has an interaction to both 'Sabbir Khan' in the document title and to 'he' in the document itself.

Note: If two spans refer to the same object/entity but with different surface forms, assign a synonym interaction between them, e.g. Example 4 in Examples FEVER – 'Oscar Isaac' is related to 'Oscar Hernandez' and to 'Oscar Isaac Hernandez'. Note: Always make sure that the annotated relations are coherent with the label provided for the instance.

## B.7 Low-level Text Spans

Annotate interactions between low-level matching text spans, inside the high-level spans, in the premise and the hypothesis. Choose a text span $\alpha$ in the premise that can be mapped to a text span $\beta$ in the hypothesis by one of the interactions: synonym, antonym, hypernym. The spans should be selected at the lowest possible level of the syntax tree, i.e. shortest possible chunks that can hold one such interaction. Do not annotate exact surface forms as synonyms, e.g. in the high-level synonym interaction "while holding to go packages" in the premise and "while holding to go packages" in the hypothesis, do not annotate the matching separate words as synonyms. Annotate as synonyms only spans that do not match in surface form. In high-level hypernym interactions, if additional details are being added in either part, leave these parts unannotated at the low level. E.g., 'holding to go packages' and 'holding packages', there is an additional modifier 'to go' added to the premise, making it more specific, thus contributing for Hypernym-h-to-p in-

teraction. Do not annotate articles. Annotate the two text spans $\alpha$ and $\beta$ as pleaf and hleaf. Connect the chunks in the interaction with an arrow as above. Annotate the type of the interaction.

Note: It is possible the high-level and low-level spans overlap in one part of the input. In such cases, annotate it both as high-level and low-level (leaf) span.

## C  Detailed Results for Human and Model Explanation

Table 8 shows the most (least for **AOPC-Suff**) important interactions according to different metrics and the rank of the most relevant (according to humans) interactions. The colors green, yellow and red indicate whether the most relevant interaction is the first (last for **AOPC-Suff**), in top (bottom for **AOPC-Suff**) 50% or bottom (**AOPC-Suff**) 50% of all interactions. Table 8 is a summary of Table 9 and Table 5 in §2 is a summary of Table 8. **AOPC-Suff** and **PHA** scores for different interactions are shown in Fig. 7 and Fig. 8, respectively.

## D  Louvain Community Detection

In an unweighted undirected graph, if the number of communities is known apriori, the minimum cut approach tells us to divide the vertices such that the number of edges between the partitions is minimized. However, that number is often unknown, and without any such constraint, this minimization would simply produce the entire graph as a single community which is not desirable.

An effective way to partition a network into communities is not just characterized by having a low number of connections between a set of vertices but rather determined by a lower number of inter-community (equivalently, higher number of intra-community) connections than what would be **expected**. The concept that the genuine community structure in a network aligns with a statistically unexpected distribution of connections can be measured through a metric called modularity (Newman and Girvan, 2004). Modularity, with a scaling factor, represents the difference between the number of edges within groups and the expected number of edges in a comparable network where connections are randomly distributed.

Louvain Community Detection algorithm (Blondel et al., 2008) uses modularity optimization to generate communities in directed graphs such as ours. The algorithm starts with each node in its

| Level | Metric | Top Relation | Relevant relation rank |
|---|---|---|---|
| **SNLI-Contradiction** | | | |
| Low | **AOPC** Comp | Antonym | 1/6 |
| | **AOPC** Suff | Antonym | 6/6 |
| | PHA | Dangler-System-P1 | 3/6 |
| High | **AOPC** Comp | Antonym | 1/8 |
| | **AOPC** Suff | Antonym | 8/8 |
| | PHA | Dangler-System-P2 | 2/8 |
| **SNLI-Entailment** | | | |
| Low | **AOPC** Comp | Dangler-System-P2 | 2/4 |
| | **AOPC** Suff | Synonym | 4/4 |
| | PHA | Synonym | 1/4 |
| High | **AOPC** Comp | Part-phrase | 5/6 |
| | **AOPC** Suff | Hypernym-P2-P1 | 6/6 |
| | PHA | Hypernym-P2-P1 | 1/6 |
| **SNLI-Neutral** | | | |
| Low | **AOPC** Comp | Hypernym-P1-P2 | 1/5 |
| | **AOPC** Suff | Hypernym-P1-P2 | 5/5 |
| | PHA | Dangler-System-P2 | 1/5 |
| High | **AOPC** Comp | Hypernym-P1-P2 | 1/7 |
| | **AOPC** Suff | Hypernym-P1-P2 | 7/7 |
| | PHA | Dangler-System-P2 | 1/7 |
| **Fever-Refutes** | | | |
| Low | **AOPC** Comp | Antonym | 1/6 |
| | **AOPC** Suff | Hypernym-P1-P2 | 5/6 |
| | PHA | Hypernym-P1-P2 | 5/6 |
| High | **AOPC** Comp | Part-phrase | 3/8 |
| | **AOPC** Suff | Hypernym-P1-P2 | 5/8 |
| | PHA | Dangler-System-P2 | 3/8 |
| **Fever-Supports** | | | |
| Low | **AOPC** Comp | Hypernym-P2-P1 | 1/4 |
| | **AOPC** Suff | Synonym | 4/4 |
| | PHA | Synonym | 1/4 |
| High | **AOPC** Comp | Part-phrase | 3/6 |
| | **AOPC** Suff | Synonym | 6/6 |
| | PHA | Dangler-System-P2 | 2/6 |
| **Fever-NEI** | | | |
| Low | **AOPC** Comp | Hypernym-P1-P2 | 1/5 |
| | **AOPC** Suff | Dangler-System-P1 | 5/5 |
| | PHA | Dangler-System-P2 | 1/5 |
| High | **AOPC** Comp | Part-phrase | 2/7 |
| | **AOPC** Suff | Random-phrase | 5/7 |
| | PHA | Dangler-System-P2 | 1/7 |

**Table 8:** Most (least for **AOPC-Suff**) important relations according to different metrics and the rank of the most relevant (according to humans) relations. The colors green, yellow and red indicate whether the most relevant relation is the first, in top (bottom for **AOPC-Suff**) 50% or bottom (top for **AOPC-Suff**) 50% of all relations.

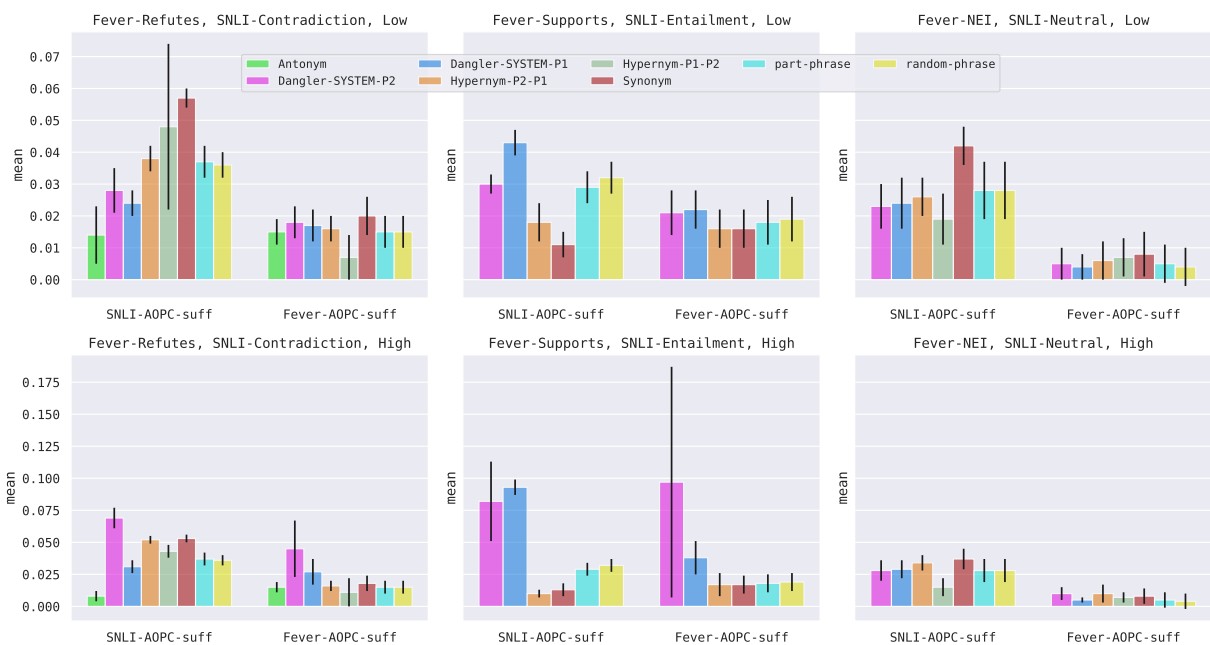

**Figure 7: AOPC-Suff** scores for different interactions: averaged (error bars: standard deviation) over annotators and models. A **lower AOPC** value for relevant interaction (Antonym for Contradiction) indicates better human-model explanation alignment.

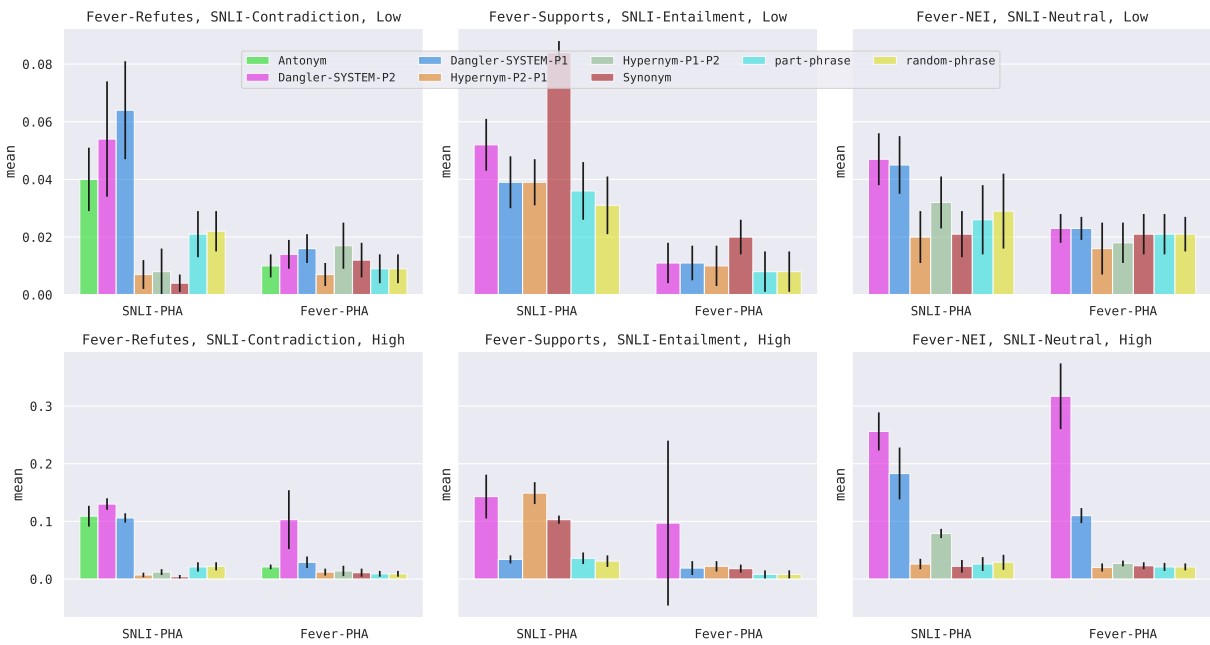

**Figure 8: PHA** scores for different interactions: averaged (error bars: standard deviation) over annotators and models. A **higher PHA** value for relevant interaction (Antonym for Contradiction) indicates better human-model explanation alignment.

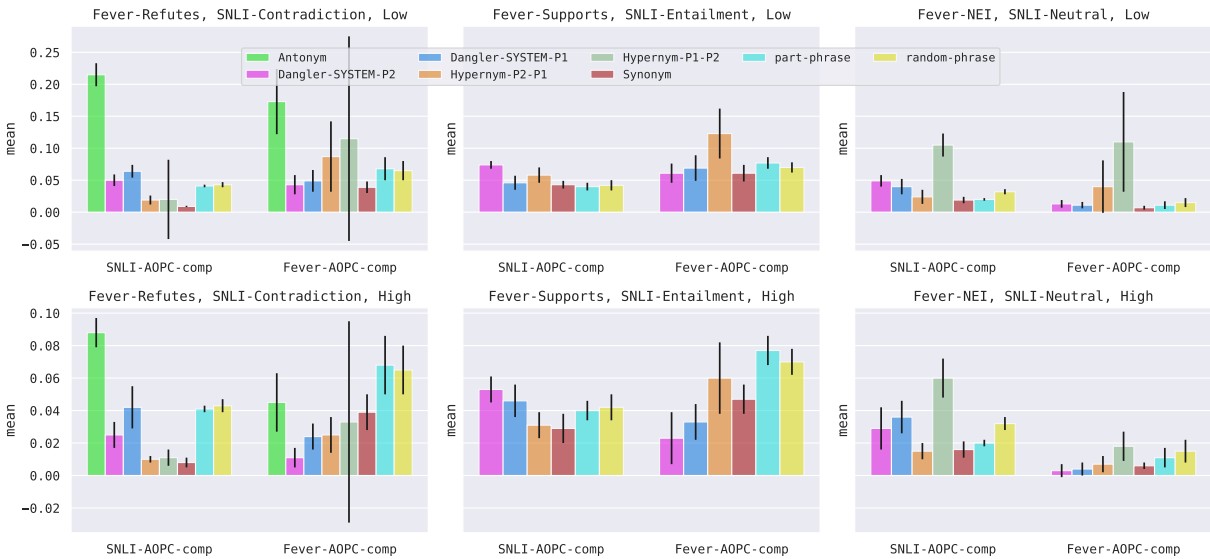

**Figure 9: AOPC-Comp** scores for different interactions: averaged (error bars: standard deviation) over annotators and models. A **higher AOPC** value for relevant interaction (Antonym for Contradiction) indicates better human-model explanation alignment. For the Part Phrase baseline, the random tokens are chosen from Part 1.

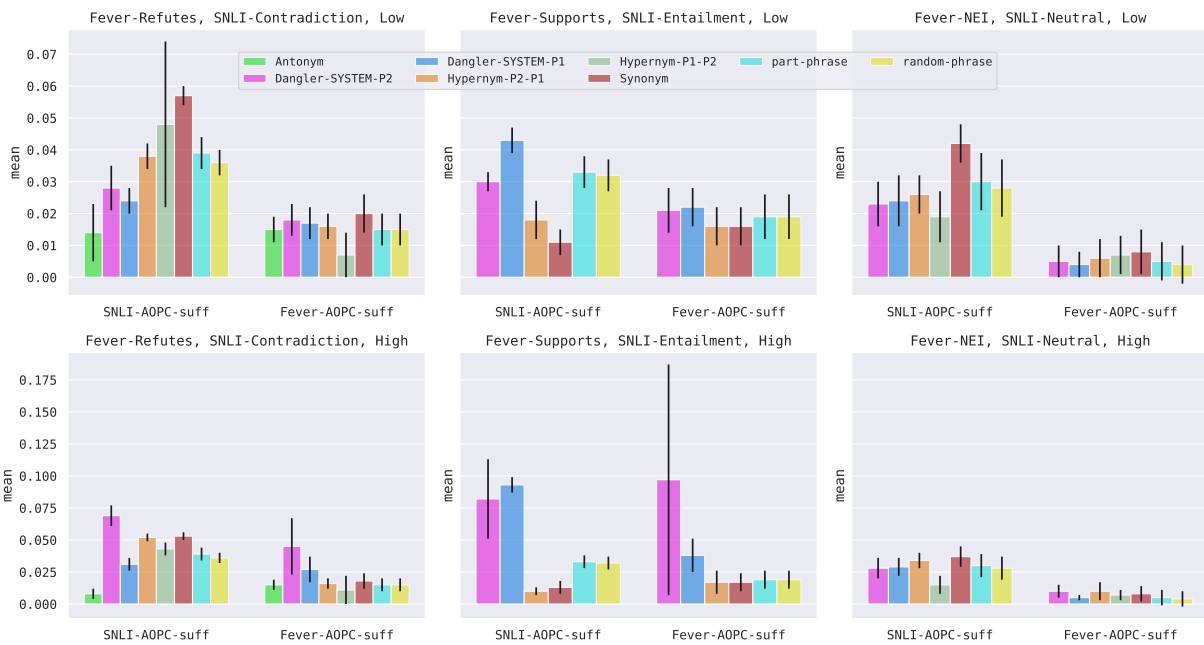

**Figure 10: AOPC-Suff** scores for different interactions: averaged (error bars: standard deviation) over annotators and models. A **lower AOPC** value for relevant interaction (Antonym for Contradiction) indicates better human-model explanation alignment. For the Part Phrase baseline, the random tokens are chosen from Part 1.

| | Dataset | Label | Relation | Level | AOPC Comp | | AOPC Suff | | PHA | |
|---|---------|-------|----------|-------|------|-----|------|-----|------|-----|
| | | | | | mean | std | mean | std | mean | std |
| 0 | FEVER | NEI | Dangler-System-P2 | High | 0.003 | 0.004 | 0.01 | 0.005 | 0.317 | 0.057 |
| 1 | FEVER | NEI | Dangler-System-P2 | Low | 0.013 | 0.006 | 0.005 | 0.005 | 0.023 | 0.005 |
| 2 | FEVER | NEI | Dangler-System-P1 | High | 0.004 | 0.004 | 0.005 | 0.002 | 0.11 | 0.013 |
| 3 | FEVER | NEI | Dangler-System-P1 | Low | 0.011 | 0.005 | 0.004 | 0.004 | 0.023 | 0.004 |
| 4 | FEVER | NEI | Hypernym-P2-P1 | High | 0.007 | 0.005 | 0.01 | 0.007 | 0.02 | 0.007 |
| 5 | FEVER | NEI | Hypernym-P2-P1 | Low | 0.04 | 0.041 | 0.006 | 0.006 | 0.016 | 0.009 |
| 6 | FEVER | NEI | Hypernym-P1-P2 | High | 0.018 | 0.009 | 0.007 | 0.004 | 0.027 | 0.005 |
| 7 | FEVER | NEI | Hypernym-P1-P2 | Low | 0.11 | 0.078 | 0.007 | 0.006 | 0.018 | 0.007 |
| 8 | FEVER | NEI | Part-phrase | High | 0.019 | 0.011 | 0.005 | 0.006 | 0.021 | 0.007 |
| 9 | FEVER | NEI | Random-phrase | High | 0.015 | 0.007 | 0.004 | 0.006 | 0.021 | 0.006 |
| 10 | FEVER | NEI | Synonym | High | 0.006 | 0.002 | 0.008 | 0.006 | 0.023 | 0.006 |
| 11 | FEVER | NEI | Synonym | Low | 0.007 | 0.003 | 0.008 | 0.007 | 0.021 | 0.007 |
| 12 | FEVER | Refutes | Antonym | High | 0.045 | 0.018 | 0.015 | 0.004 | 0.021 | 0.004 |
| 13 | FEVER | Refutes | Antonym | Low | 0.173 | 0.051 | 0.015 | 0.004 | 0.01 | 0.004 |
| 14 | FEVER | Refutes | Dangler-System-P2 | High | 0.011 | 0.006 | 0.045 | 0.022 | 0.103 | 0.051 |
| 15 | FEVER | Refutes | Dangler-System-P2 | Low | 0.043 | 0.015 | 0.018 | 0.005 | 0.014 | 0.005 |
| 16 | FEVER | Refutes | Dangler-System-P1 | High | 0.024 | 0.008 | 0.027 | 0.01 | 0.029 | 0.01 |
| 17 | FEVER | Refutes | Dangler-System-P1 | Low | 0.049 | 0.017 | 0.017 | 0.005 | 0.016 | 0.005 |
| 18 | FEVER | Refutes | Hypernym-P2-P1 | High | 0.025 | 0.011 | 0.016 | 0.004 | 0.012 | 0.006 |
| 19 | FEVER | Refutes | Hypernym-P2-P1 | Low | 0.087 | 0.055 | 0.016 | 0.004 | 0.007 | 0.004 |
| 20 | FEVER | Refutes | Hypernym-P1-P2 | High | 0.033 | 0.062 | 0.011 | 0.011 | 0.014 | 0.009 |
| 21 | FEVER | Refutes | Hypernym-P1-P2 | Low | 0.115 | 0.16 | 0.007 | 0.007 | 0.017 | 0.008 |
| 22 | FEVER | Refutes | Part-phrase | High | 0.08 | 0.019 | 0.015 | 0.005 | 0.009 | 0.005 |
| 23 | FEVER | Refutes | Random-phrase | High | 0.065 | 0.015 | 0.015 | 0.005 | 0.009 | 0.005 |
| 24 | FEVER | Refutes | Synonym | High | 0.039 | 0.011 | 0.018 | 0.006 | 0.011 | 0.007 |
| 25 | FEVER | Refutes | Synonym | Low | 0.039 | 0.009 | 0.02 | 0.006 | 0.012 | 0.006 |
| 26 | FEVER | Supports | Dangler-System-P2 | High | 0.023 | 0.016 | 0.097 | 0.09 | 0.097 | 0.143 |
| 27 | FEVER | Supports | Dangler-System-P2 | Low | 0.061 | 0.015 | 0.021 | 0.007 | 0.011 | 0.007 |
| 28 | FEVER | Supports | Dangler-System-P1 | High | 0.033 | 0.011 | 0.038 | 0.013 | 0.019 | 0.012 |
| 29 | FEVER | Supports | Dangler-System-P1 | Low | 0.069 | 0.02 | 0.022 | 0.006 | 0.011 | 0.006 |
| 30 | FEVER | Supports | Hypernym-P2-P1 | High | 0.06 | 0.022 | 0.017 | 0.009 | 0.022 | 0.009 |
| 31 | FEVER | Supports | Hypernym-P2-P1 | Low | 0.123 | 0.039 | 0.016 | 0.006 | 0.01 | 0.007 |
| 32 | FEVER | Supports | Part-phrase | High | 0.095 | 0.004 | 0.018 | 0.007 | 0.008 | 0.007 |
| 33 | FEVER | Supports | Random-phrase | High | 0.07 | 0.008 | 0.019 | 0.007 | 0.008 | 0.007 |
| 34 | FEVER | Supports | Synonym | High | 0.047 | 0.009 | 0.017 | 0.007 | 0.018 | 0.007 |
| 35 | FEVER | Supports | Synonym | Low | 0.061 | 0.013 | 0.016 | 0.006 | 0.02 | 0.006 |
| 36 | SNLI | Contradiction | Antonym | High | 0.088 | 0.009 | 0.008 | 0.004 | 0.109 | 0.018 |
| 37 | SNLI | Contradiction | Antonym | Low | 0.215 | 0.018 | 0.014 | 0.009 | 0.04 | 0.011 |
| 38 | SNLI | Contradiction | Dangler-System-P2 | High | 0.025 | 0.008 | 0.069 | 0.008 | 0.13 | 0.01 |
| 39 | SNLI | Contradiction | Dangler-System-P2 | Low | 0.05 | 0.009 | 0.028 | 0.007 | 0.054 | 0.02 |
| 40 | SNLI | Contradiction | Dangler-System-P1 | High | 0.042 | 0.013 | 0.031 | 0.005 | 0.106 | 0.008 |
| 41 | SNLI | Contradiction | Dangler-System-P1 | Low | 0.064 | 0.01 | 0.024 | 0.004 | 0.064 | 0.017 |
| 42 | SNLI | Contradiction | Hypernym-P2-P1 | High | 0.01 | 0.002 | 0.052 | 0.003 | 0.007 | 0.004 |
| 43 | SNLI | Contradiction | Hypernym-P2-P1 | Low | 0.019 | 0.007 | 0.038 | 0.004 | 0.007 | 0.005 |
| 44 | SNLI | Contradiction | Hypernym-P1-P2 | High | 0.011 | 0.005 | 0.043 | 0.005 | 0.012 | 0.005 |
| 45 | SNLI | Contradiction | Hypernym-P1-P2 | Low | 0.02 | 0.062 | 0.048 | 0.026 | 0.008 | 0.008 |
| 46 | SNLI | Contradiction | Part-phrase | High | 0.049 | 0.006 | 0.037 | 0.005 | 0.021 | 0.008 |
| 47 | SNLI | Contradiction | Random-phrase | High | 0.043 | 0.004 | 0.036 | 0.005 | 0.023 | 0.007 |
| 48 | SNLI | Contradiction | Synonym | High | 0.008 | 0.003 | 0.053 | 0.003 | 0.004 | 0.003 |
| 49 | SNLI | Contradiction | Synonym | Low | 0.009 | 0.001 | 0.057 | 0.003 | 0.004 | 0.003 |
| 50 | SNLI | Entailment | Dangler-System-P2 | High | 0.053 | 0.008 | 0.082 | 0.031 | 0.143 | 0.038 |
| 51 | SNLI | Entailment | Dangler-System-P2 | Low | 0.074 | 0.006 | 0.03 | 0.003 | 0.052 | 0.009 |
| 52 | SNLI | Entailment | Dangler-System-P1 | High | 0.046 | 0.01 | 0.093 | 0.006 | 0.034 | 0.007 |
| 53 | SNLI | Entailment | Dangler-System-P1 | Low | 0.046 | 0.011 | 0.043 | 0.004 | 0.039 | 0.009 |
| 54 | SNLI | Entailment | Hypernym-P2-P1 | High | 0.031 | 0.008 | 0.01 | 0.003 | 0.149 | 0.019 |
| 55 | SNLI | Entailment | Hypernym-P2-P1 | Low | 0.058 | 0.012 | 0.018 | 0.006 | 0.039 | 0.008 |
| 56 | SNLI | Entailment | Part-phrase | High | 0.076 | 0.015 | 0.029 | 0.005 | 0.036 | 0.01 |
| 57 | SNLI | Entailment | Random-phrase | High | 0.042 | 0.008 | 0.032 | 0.005 | 0.031 | 0.01 |
| 58 | SNLI | Entailment | Synonym | High | 0.029 | 0.009 | 0.013 | 0.005 | 0.103 | 0.007 |
| 59 | SNLI | Entailment | Synonym | Low | 0.043 | 0.006 | 0.011 | 0.004 | 0.084 | 0.004 |
| 60 | SNLI | Neutral | Dangler-System-P2 | High | 0.029 | 0.013 | 0.028 | 0.008 | 0.256 | 0.033 |
| 61 | SNLI | Neutral | Dangler-System-P2 | Low | 0.049 | 0.009 | 0.023 | 0.007 | 0.047 | 0.009 |
| 62 | SNLI | Neutral | Dangler-System-P1 | High | 0.036 | 0.01 | 0.029 | 0.007 | 0.183 | 0.045 |
| 63 | SNLI | Neutral | Dangler-System-P1 | Low | 0.04 | 0.012 | 0.024 | 0.008 | 0.045 | 0.01 |
| 64 | SNLI | Neutral | Hypernym-P2-P1 | High | 0.015 | 0.005 | 0.034 | 0.006 | 0.026 | 0.009 |
| 65 | SNLI | Neutral | Hypernym-P2-P1 | Low | 0.024 | 0.011 | 0.026 | 0.006 | 0.02 | 0.009 |
| 66 | SNLI | Neutral | Hypernym-P1-P2 | High | 0.06 | 0.012 | 0.015 | 0.007 | 0.079 | 0.008 |
| 67 | SNLI | Neutral | Hypernym-P1-P2 | Low | 0.105 | 0.018 | 0.019 | 0.008 | 0.032 | 0.009 |
| 68 | SNLI | Neutral | Part-phrase | High | 0.036 | 0.01 | 0.028 | 0.009 | 0.026 | 0.012 |
| 69 | SNLI | Neutral | Random-phrase | High | 0.032 | 0.004 | 0.028 | 0.009 | 0.029 | 0.013 |
| 70 | SNLI | Neutral | Synonym | High | 0.016 | 0.005 | 0.037 | 0.008 | 0.022 | 0.011 |
| 71 | SNLI | Neutral | Synonym | Low | 0.019 | 0.005 | 0.042 | 0.006 | 0.021 | 0.008 |

**Table 9:** **AOPC** comprehensiveness, **AOPC** sufficiency and **PHA** scores for FEVER and SNLI across different labels, relations, and levels.

community and iteratively moves them to the neighboring communities if that contributes to a positive modularity gain. The modularity gain of moving a node $i$ into a community $C$ can be summarized as $\Delta Q = \frac{k_{i,in}}{m} - \frac{k_i^{out} \cdot \Sigma_{tot}^{in} + k_i^{in} \cdot \Sigma_{tot}^{out}}{m^2}$ where $k_i^{out}$, $k_i^{in}$ are the outer and inner weighted degrees of node

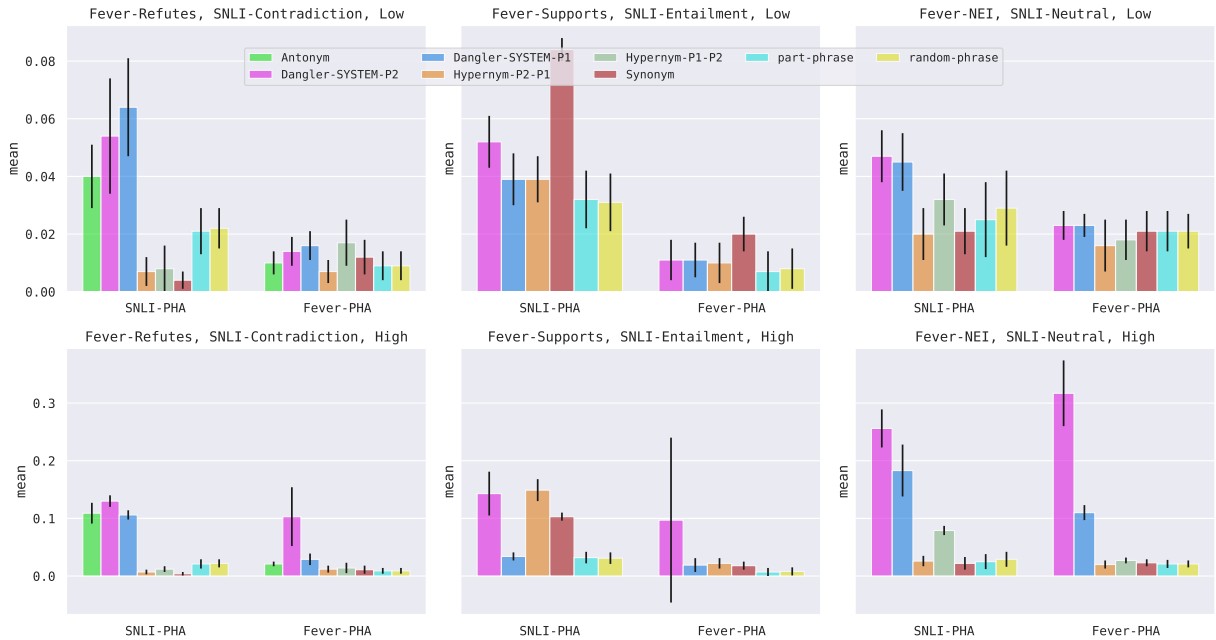

**Figure 11: PHA** scores for different interactions: averaged (error bars: standard deviation) over annotators and models. A **higher PHA** value for relevant interaction (Antonym for Contradiction) indicates better human-model explanation alignment. For the Part Phrase baseline, the random tokens are chosen from the Part 1.

$i$, $\Sigma_{tot}^{in}$, $\Sigma_{tot}^{out}$ are the sum of in-going and out-going links incident to nodes in $C$. The algorithm terminates when no such gain can be achieved.

# E    Explanation Extraction Results

Top-1 and Top-3 evaluation results are shown in Fig. 12 and Fig. 13, respectively. See Table 10 for some examples of top-1 explanations generated by our method.

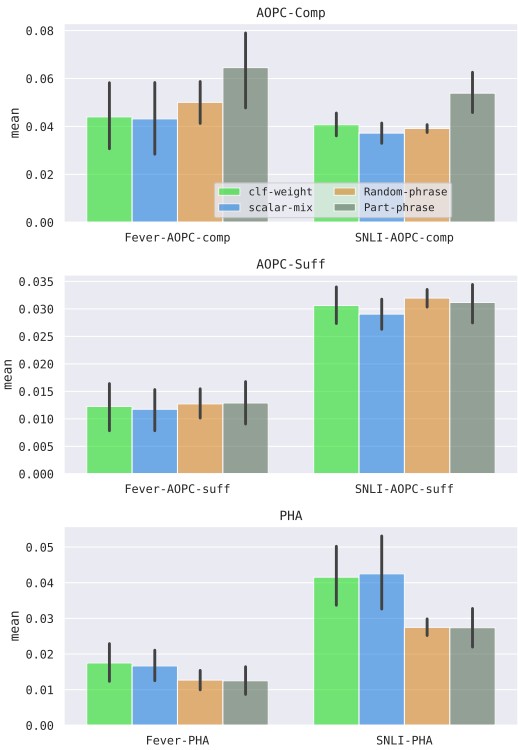

**Figure 12:** Top-1 evaluation scores for Louvain community detection over two types of attention graphs, along with the Part Phrase and Random Phrase baselines. **AOPC-Comp** & **PHA**: higher is better, **AOPC-Suff**: lower is better.

| Label | Part 1 | Part 2 | Top-1 explanation | Comment |
|---|---|---|---|---|
| Contradiction | One tan girl with a wool hat is running and leaning over an object, while another person in a wool hat is sitting on the ground. | A boy runs into a wall | (One tan girl, a boy) | correct, a tan girl is an antonym to a boy |
| | A young family enjoys feeling ocean waves lap at their feet. | A family is out at a restaurant. | (feeling, is) | incorrect, these are not antonym relations. |
| Entailment | A young family enjoys feeling ocean waves lap at their feet. | A family is at the beach. | (ocean waves, beach) | correct, ocean waves indicate beach, i.e., a synonym relation |
| | A couple walk hand in hand down a street. | A couple is walking together. | (hand, together) | incorrect, there is no relation. |
| Neutral | A couple walk hand in hand down a street. | The couple is married. | (A couple, married) | correct, hypernym-P1-P2 as "a couple" does not necessarily imply married, but the reverse is true, |
| | One tan girl with a wool hat is running and leaning over an object, while another person in a wool hat is sitting on the ground. | A man watches his daughter leap | (girl, daughter) | Correct, hypernym-P1-P2 as "girl" does not necessarily imply daughter, but the reverse is true. |
| | One tan girl with a wool hat is running and leaning over an object, while another person in a wool hat is sitting on the ground. | A man watches his daughter leap | (while, watches) | Incorrect, there is no relation. |

**Table 10:** Example top-1 explanations generated on SNLI by the **Classifer-Weight** method.

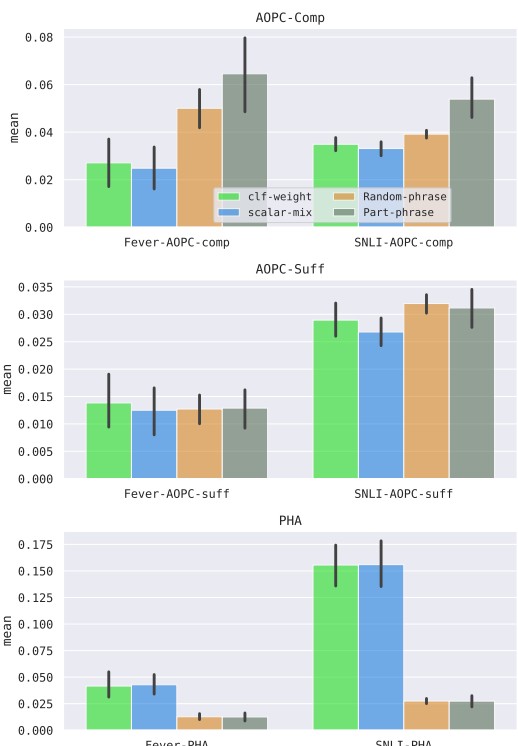

**Figure 13:** Top-5 evaluation scores for Louvain community detection over two types of attention graphs, along with the Part Phrase and Random Phrase baselines. **AOPC-Comp** & **PHA**: higher is better, **AOPC-Suff**: lower is better.