# OpenReview forum: "Explaining Interactions Between Text Spans"
_EMNLP/2023/Conference — EMNLP 2023 Main_

### Official Review · Reviewer_yM6C · 2023-07-29

**Soundness:** 4

**Excitement:**

4: Strong: This paper deepens the understanding of some phenomenon or lowers the barriers to an existing research direction.

**Justification For Ethical Concerns:**

I do not see ethical concerns in this paper.

**Missing References:**

I found some prior works about span-level interaction. It might be nice to add them to the related works.
- Lu et al., (2021) Learning Span-Level Interactions for Aspect Sentiment Triplet Extraction
- Dixit and Al-Onaizan (2019) Span-Level Model for Relation Extraction
- Gong et al., (2017) Natural Language Inference over Interaction Space

**Paper Topic And Main Contributions:**

This paper considers a crucial step in the reasoning process over spans of tokens from different parts of the input: span interactions. There is a lack of explainability technique that unveils this step in the automatic solving of questions. There is also a lack of annotated ground truths.

To address these research gaps, this paper collects a dataset, SpanEx, containing 7000+ instances annotated for span interactions. This is the first dataset with human phrase-level interaction explanations with explicit labels for interaction types. The human annotators often agree on the interactions but provide complementary reasons for a prediction.

Further, this paper tests if the fine-tuned LLMs follow similar decision-making procedure (in terms of reasoning about the span-level interactions) as humans. With the sufficiency and comprehensiveness scores, this paper finds that the models rely on interactions that are consistent with the human annotations (and the consistency is higher when the humans agree more with each other).

To add to the utility of the findings, this paper propose a novel approach for generating interaction explanations that connect textual spans from different parts of the input. This approach is based on community structure detection algorithms (the Louvain algorithm).

**Questions For The Authors:**

Could you comment on the differences between the high-level and low-level spans?

**Reasons To Accept:**

- The span-level interaction is an important step in the decision-making procedure that deserves more attention. This paper provides an annotated dataset and tools that facilitate the research towards this direction.
- This paper provides interesting findings about how some LLMs respond to the span-level interactions.

**Reasons To Reject:**

- The "high-level vs low-level" spans appear relative. For example, "video games" is considered as low-level when compared to "54 video games", yet it is high-level compared to "games". At a second thought, shouldn't "video games", which has an additional modifier that constrains the scope, be more low-level than "games"? But this point appears minor compared to the core contribution of this paper.
- Another minor weakness: The models in 3.2 (BERT and RoBERTa with different sizes) are all bidirectional models. It is unsure whether the findings can generalize to uni-directional models like GPT.

**Reproducibility:**

4: Could mostly reproduce the results, but there may be some variation because of sample variance or minor variations in their interpretation of the protocol or method.

**Reviewer Confidence:**

3: Pretty sure, but there's a chance I missed something. Although I have a good feel for this area in general, I did not carefully check the paper's details, e.g., the math, experimental design, or novelty.

**Typos Grammar Style And Presentation Improvements:**

- Line 356: 4 out of 6 is not 70%. It is 66.7%, if you want to keep the number of precision points consistent throughout the paper (e.g., in lines 419-420 you took three precision points).

---

> ### Author Rebuttal · Authors · 2023-08-28
>
> Thank you for appreciating the novelty and soundness of our work. Our responses to the comments are given below.
>
> -  _Could you comment on the differences between the high-level and low-level spans?_
>
> A high-level span is the largest contiguous sequence of tokens that a) is _not_ the whole part, (i.e., not the entire premise or hypothesis) b) bears meaning in itself, and c) can be associated with a span from another part using one of the defined relations.
>
> Consider the premise “Two women are running” and the hypothesis “Two men are walking”. The label is ''contradiction’’.
>
> The reasoning behind the "contradiction" decision needs to be established through the constituents in the sentences. We see that the subjects, made out of noun phrases are antonyms: "two men" (premise) and "two women’’ (hypothesis), and so are the predicates, made out of verb phrases: "are running" and "are walking". These are the largest meaningful constituents where such relationships can be established. Therefore, they are considered "high-level" spans.
>
> A low-level span is the smallest meaning-bearing constituent/text that still holds a relation. For the given example, these would be "man"/"woman", "running"/"walking".
>
> For annotating high-level span boundaries, the annotators were shown the constituency parse tree as a suggestion, but it was not enforced that the boundaries _must_ adhere to the constituents, as the semantic segmentation of a sentence does not always adhere to the syntactic one.
>
>  - _The "high-level vs low-level" spans appear relative. For example, "video games" is considered as low-level when compared to "54 video games", yet it is high-level compared to "games". At a second thought, shouldn't "video games", which has an additional modifier that constrains the scope, be more low-level than "games"? But this point appears minor compared to the core contribution of this paper._
>
>  We have clarified the concept of the high vs. low-level spans above. One final thing to note is that we consider a longer sequence as high-level and a shorter one, within the longer sequence, as low-level. These were not designed to represent hypernym/hyponym relations between the high-level and low-level span, i.e., they are not dependent on the constraining scope.
>
> -  _Another minor weakness: The models in 3.2 (BERT and RoBERTa with different sizes) are all bidirectional models. It is unsure whether the findings can generalize to uni-directional models like GPT._
>
> We have indeed analyzed the inner workings of the most popular bidirectional Transformer models. With our dataset and the performed analysis, we have set the ground and only scratched the surface of the prospects to inspect the inner workings of a multitude of different architectures for span interactions, such as auto-regressive Transformer models. The implementation (released upon acceptance) can be easily adapted to perform the following studies in future work.

---

### Official Review · Reviewer_vXE4 · 2023-08-04

**Typos Grammar Style And Presentation Improvements:** 1. The notations in Section 3.1 may b…
**Soundness:** 3

**Excitement:**

3: Ambivalent: It has merits (e.g., it reports state-of-the-art results, the idea is nice), but there are key weaknesses (e.g., it describes incremental work), and it can significantly benefit from another round of revision. However, I won't object to accepting it if my co-reviewers champion it.

**Missing References:**

The following references are the works that are mentioned in this review.

[1] Jiang, Zhongtao, et al. "Alignment rationale for natural language inference." Proceedings of the 59th Annual Meeting of the Association for Computational Linguistics and the 11th International Joint Conference on Natural Language Processing (Volume 1: Long Papers). 2021.

[2] Stacey, Joe, et al. "Logical Reasoning with Span-Level Predictions for Interpretable and Robust NLI Models." Proceedings of the 2022 Conference on Empirical Methods in Natural Language Processing. 2022.

[3] Wu, Zijun, et al. "Weakly supervised explainable phrasal reasoning with neural fuzzy logic." arXiv preprint arXiv:2109.08927 (2021).

[4] Kim, Youngwoo, Razieh Rahimi, and James Allan. "Alignment Rationale for Query-Document Relevance." Proceedings of the 45th International ACM SIGIR Conference on Research and Development in Information Retrieval. 2022.

**Paper Topic And Main Contributions:**

This paper introduces the SpanEx dataset, which contains human annotations that illustrate how text spans in NLU datasets interact with each other to reach a final classification decision.
Specifically, the natural language inference dataset SNLI and the fact-checking dataset FEVER have been annotated. Pairs of spans from two sentences are aligned and labeled as either synonym, hypernym, or antonym. Experiments demonstrate the agreement between the SpanEx dataset and the model, based on changes in model predictions after features are removed.
Moreover, the paper proposes a novel community detection-based unsupervised method to extract interaction explanations.

**Reasons To Accept:**

1. This paper introduces an interesting, novel dataset that contains annotations of span interactions. This dataset can serve as an invaluable resource for evaluating interaction explanations in complex LLM-based NLU systems.
2. The dataset boasts high inter-annotator agreement.
3. The example in Figure 1 provides a clear sense of what has been annotated.

**Reasons To Reject:**

1. Details about the dataset construction are either not provided or difficult to understand. Specifically, descriptions regarding the criteria for segmenting spans, the implications of high-level and low-level spans, and the concept of relaxed span match are unclear.
2. The experiments mainly use feature (token*) selection and removal to assess annotation quality. However, these evaluation techniques may not be ideal for comparing span interaction explanations. Consequently, the findings from Section 3 should be carefully examined.


* I think that if the synonym or hypernym span pairs are deleted, it is more probable that the classification decisions will be preserved. This would lead to a low AOPC-Comp score. Figure 2 also shows lower AOPC-Comp scores for synonym. This contradicts the paper’s argument in Line 313 which suggests that a higher AOPC-Comp score is desirable.
* However, even a small AOPC-Comp score is not effective measure for important interactions, because selecting two unimportant spans might also result in small score changes.
* To simulate removing only the interaction features and not the span themselves, removing attention of Transformer architecture could be the better evaluation [1], or conditionally removing tokens [4].

**Reproducibility:**

3: Could reproduce the results with some difficulty. The settings of parameters are underspecified or subjectively determined; the training/evaluation data are not widely available.

**Reviewer Confidence:**

3: Pretty sure, but there's a chance I missed something. Although I have a good feel for this area in general, I did not carefully check the paper's details, e.g., the math, experimental design, or novelty.

---

> ### Author Rebuttal · Authors · 2023-08-29
>
> Thank you for appreciating the novelty of the proposed dataset and its value for evaluating interaction explanations. Our responses to the comments are below.
>
> ### Reasons to reject:
>
> _1. Details about the dataset construction are either not provided or difficult to understand. Specifically, descriptions regarding the criteria for segmenting spans, the implications of high-level and low-level spans, and the concept of relaxed span match are unclear._
>
> In addition to the dataset construction description in Section 2, we have also provided the complete annotation guidelines in the appendix (as pointed out in L192) and plan to release the full implementations upon acceptance. We now provide additional details about the pointed aspects as follows. We will include these in the main manuscript as well.
>
> __Criteria for segmenting spans, the implications of high-level and low-level spans.__: A high-level span is the largest contiguous sequence of tokens that a) is _not_ the whole part, (i.e., not the entire premise or hypothesis) b) bears meaning in itself, and c) can be associated with a span from another part using one of the defined relations.
>
> Consider the premise “Two women are running” and the hypothesis “Two men are walking”. The label is "contradiction’’.
>
> The reasoning behind the "contradiction" decision needs to be established through the constituents in the sentences. We see that the subjects, made out of noun phrases are antonyms: "two men" (premise) and "two women" (hypothesis), and so are the predicates, made out of verb phrases: "are running" and "are walking". These are the largest meaningful constituents where such relationships can be established. Therefore, they are considered "high-level" spans.
>
> A low-level span is the smallest meaning-bearing constituent/text that still holds a relation. For the given example, these would be "man"/"woman’",  "running’’/"walking’’.
>
> For annotating high-level span boundaries, the annotators were shown the constituency parse tree as a suggestion, but it was not enforced that the boundaries _must_ adhere to the constituents, as the semantic segmentation of a sentence does not always adhere to the syntactic one.
>
> __The concept of relaxed span match.__: In the relaxed span agreement, two annotated interactions "match" if there is at least one common token common between them, as pointed out in Table 3. The motivation is to _not_ penalize small boundary mismatches, as explained in L226-230. We will be sure to clarify this further should the paper get accepted.
>
> _2. The experiments mainly use feature (token*) selection/removal as means to evaluate the annotations quality. I disagree that this is a valid measure for span interaction. This harms implications of Section 3._
>
> We want to point out that in our work a) an annotation constitutes an interaction explanation, and b) we adopt an evaluation protocol established in prior works (L266) on interaction explanations. While there are prospects for finding a more suitable evaluation protocol in future work, this is not the focus of this work.
>
> _-   I think that if the synonym or hypernym span pairs are deleted, it is more probable that the classification decisions will be preserved. This would lead to a low AOPC-Comp score. Figure 2 also shows lower AOPC-Comp scores for synonym. This contradicts the paper’s argument in Line 313 which suggests that a higher AOPC-Comp score is desirable._
>
> As pointed out in the manuscript, a higher AOPC-Comp score is indeed desirable, as it indicates a more faithful explanation. However, we note that Synonym/Hypernym-P2-P1 relations are important for instances with the Entailment/Supports labels. For other labels, e.g., contradiction, these relations are less relevant than other relations, say, Antonym. Consequently, the AOPC-comp scores for Synonym interactions should be lower as they are in Figure 2. Figure 2 and Table 4 further confirm that in most cases, the relevant relations indeed have higher AOPC-Comp scores than the non-relevant ones. We do agree that the Synonym relations are less important even for Entailment labels. __However, we believe that is a finding, not a limitation.__
>
> _-   However, even a small AOPC-Comp score is not effective measure for important interactions, because selecting two unimportant spans might also result in small score changes._
>
> That is true, selecting two unimportant spans would not significantly change the AOPC-Comp score, which is desirable. This does not seem to be a limitation.
>
> _-   To simulate removing only the interaction features and not the span themselves, removing attention of Transformer architecture could be the better evaluation [1], or conditionally removing tokens [4].__
>
> Thank you for pointing out the resources [1] and [4]. However, we found that the evaluation methods proposed in [1] or [4] are not different from the evaluation methods employed in our work insofar they are all token deletion/substituition based. [1] evaluates explanations using "Area Over Reservation Curve’’, which is very similar to AOPC-comp, and is a token selection/removal-based evaluation (see Eq. 13 in [1]). In [4], the evaluation metrics are again necessity (comprehensiveness) and sufficiency, based on token selection/removal: “_We use the two criteria sufficiency and necessity [1] in our metrics. Sufficiency measures whether a rationale is sufficient for a model prediction by comparing the model output for the full input to its output for the input built from the rationale. Necessity measures whether a rationale captures only the necessary information by comparing the model output when the rationale is removed._” Specifically, the alignment independent metrics (Sec. 4.1) in [4] are the same as ours if we constrain the perturbations to only one part (premise/hypothesis). The deletion-based metrics (Sec 4.2) are very similar to AOC-comp and AOC-suff. Finally, the substitution-based metrics (Sec 4.3) are not applicable to our case as that would require replacing the premise/hypothesis with pre-determnied spans. [4] does use an “attention mask’’ metric, which in our case would translate to zeroing out the attention weight between the explanation tokens. This is a) not a standard evaluation metric, and b) is proposed by [1] as a method to generate explanations. Also, [4] finds problems with this evaluation metric. In summary, both [1] and [4] _differ from us in how the explanations are generated_ but use _boradly the same protocol as ours in evaluating_ them. Nonetheless, we appreciate the pointer to [1] as that is very relevant to our work.
>
> As for the reproducibility of the paper, the complete implementation with all hyper-parameters will be released along with the proposed dataset.
>
> ### Typos Grammar Style And Presentation Improvements:
>
>  _1. The notations in Section 3.1 may be confusing because they are not clearly describe what it means._
>
> We understand the concern raised about the clarity of the notations and their descriptions. In the revised manuscript, we will focus on significantly improving the descriptions of the notations in question.
>
> _2. The definition of "feature" needs clearer elaboration. This clarity is essential, especially when interpreting descriptions such as "removing the k most important features" (as mentioned in line 287)._
>
> We used the word feature to indicate both tokens/spans as we believed this would be clear from the context, but we would clarify this further.
>
> _3. The criteria distinguishing between high and low-level spans are ambiguous. I understand that if a span is longer and contains another span it can be considered "high-level" compared to another, but I am not convinced if clear binary classification as high and low-level is possible._
>
> Please see above our elaboration on the low- and high-level spans. In addition, in our annotation protocol, an annotator chooses the high-level span first, and then the low-level span inside it. It is possible that looking just at a span one can not clearly say whether it is high or low level, but in our case, that problem is never raised by design.
>
> _4. My understanding is that a span is considered "sufficient to bear meaning" if it can be unambiguously aligned with a corresponding span in another text. However, this interpretation does not justify why 'has made' (as referenced in Figure 1) should be treated as a span, especially when 'made' by itself could also qualify as one.
> There are papers on span-level modeling of NLI tasks , and they typically decide span boundary based on syntactic borders, such as splitting sentence by noun phrases [2], or extracting noun phrases and verb phrases [3]. If authors can provide how the guideline or annotators’ tendency aligns with existing practice of sentence segmentation for NLI or similar tasks it will be helpful for reader to understand. Authors might compare these spans with literature about token chunking to justify or hints the span annotation guidelines._
>
> We have clarified the concept of high vs. low-level spans above. Importantly, we did provide the annotators with information about the constituent boundaries, but we did not enforce them. The suggestion for including the alignment between the annotated span boundaries and constituents is indeed valuable and could be addressed in future work.
>
> _5. Line 233: "Relaxed Span Match agreement, where the case where interactions are annotated by three annotators at the high level and at least two at low level" If the objective is to highlight the greater agreement among high-level spans, it might be more straightforward to report the agreement for high-level annotations and low-level annotations separately._
>
> This is reported in Table 3.
>
> _6. Line 244: I believe it should reference Table 3, not Table 7._
>
> This refers to Table 7 in the appendix, we will mention this explicitly in the revised manuscript.
>
> We express our gratitude for your thorough review and valuable insights. We appreciate your favorable comments regarding the utility of the proposed dataset for further research in interaction explanations in NLP models. We have tried our best to address the issues raised in your review and hope that clears any misunderstandings. We understand your unhappiness with the lack of exposition in section 3, and we will address that if accepted. Finally, we hope that the clarifications provided here will reaffirm the __soundness__ of our study.

---

### Official Review · Reviewer_hKS7 · 2023-08-07

**Soundness:** 4

**Excitement:**

4: Strong: This paper deepens the understanding of some phenomenon or lowers the barriers to an existing research direction.

**Paper Topic And Main Contributions:**

This paper provides a corpus of span interaction annotations for two datasets (SNLI and FEVER) that can provide insight into reasoning process that humans may use to solve the associated tasks. The main goal of this corpus is to check whether the NLP models also use similar reasoning processes (or atleast focus on similar span interactions) when reaching their own predictions for these tasks.

The corpus provides span interactions of three types between token spans of two parts of the inputs - Synonyms, Antonyms and Hypernym relations. Each example is annotated by 3 annotators and the authors report a high Inter-Annotator Agreement.

The authors then test whether existing BERT based models also use similar reasoning mechanisms to reach their conclusion. They measure the AOPC based metrics (comprehensiveness and sufficiency) as the explanation tokens are either removed or kept within the inputs. The human annotated explanations are compared to two simple baselines: random selection of token spans, and keeping human explanation for one part of the input while choosing random span for other. Overall, the results are decidedly mixed where the model follows the intuitive behavior in some cases (for example, focusing on antonyms for contradiction) but not others. The authors do a good job teasing out these behaviors.

The authors also propose an algorithm to extract span interactions that the model might be using by extracting important span pairs based on final layer attention matrices. They use community detection algorithms on bipartite graphs of tokens using attention weights as edge weights. Overall, they observe high post hoc accuracy using only the extracting span pairs compared to the baselines but AOPC scores do not seem to show significant differences.

Overall, I find the paper to be well written and authors provided analysis to support the research questions they set out for themselves. I also find the corpus to be useful for future explainability research on different parts of the input are interacting with each other to generate a prediction.

While not necessary, one additional experiment that may strengthen the paper is to train a model with provided explanations to see if it returns superior accuracy compared to current models. This training may take a simple form of an additional loss supervising the attention matrix, etc.

**Questions For The Authors:**

1. On Page 3, Footnote 3, You mention that the datasets are highly curated which should reduce false positives of surface matched synonyms. I am not sure I understand in what you meant the datasets are curated? Can you expand on this?

2. In the part phrase baseline, do you randomly select which part of the input the human explanation would be used and which one random explanation will come from?

3. On line 468, you introduce the symbol c_i but it is not clear where is comes from. Is it ith dimension of CLS vector?

4. When constructing G_I, do you only keep the attention weights between tokens that belong to different parts of the input to get the bipartite graph? Are the attention scores renormalized?

**Reasons To Accept:**

The main strengths of this paper are:
1. Collection of useful corpus of explanations that would provide research opportunities on whether the model follows similar reasoning processes as humans when making predictions on multi-part inputs.
2. Experiments show that some current models do not yet use similar reasoning mechanisms to reach their conclusions, allowing the community to further investigate what mechanisms are in fact being used by the models and how to guide them to be right for the right reasons.
3. Paper is well written and understandable by a technical reader.

**Reasons To Reject:**

I have not found specific risks that should prevent this paper from appearing at a conference.

**Reproducibility:**

3: Could reproduce the results with some difficulty. The settings of parameters are underspecified or subjectively determined; the training/evaluation data are not widely available.

**Reviewer Confidence:**

3: Pretty sure, but there's a chance I missed something. Although I have a good feel for this area in general, I did not carefully check the paper's details, e.g., the math, experimental design, or novelty.

---

> ### Author Rebuttal · Authors · 2023-08-28
>
> Thank you for appreciating the usefulness of the dataset for investigations of the models' inner workings, for guiding them to be right for the right reasons, and for future explainability research in general. Please find the answers to the questions raised below.
>
> -  _On Page 3, Footnote 3, You mention that the datasets are highly curated which should reduce false positives of surface matched synonyms. I am not sure I understand in what you meant the datasets are curated? Can you expand on this?_
>
> To reduce false positive surface matches, we removed those where both spans include only stopwords. In addition, we took care of false negative surface matches as we found that the original datasets (FEVER and SNLI) have few instances with morphological errors (e.g., spelling mistakes, missing apostrophes). We instructed annotators to include those in their annotations. The implementation of the detailed data curation will be included upon release of the dataset, accompanied by a more meticulous description of the aforementioned steps in the manuscript.
>
> -  _In the part phrase baseline, do you randomly select which part of the input the human explanation would be used and which one random explanation will come from?_
>
> We chose to report the results for the random tokens from the hypothesis part but we do have results where the random tokens were chosen from the premise part. They were not included as the results showed similar trends. We will add the new set of results in the appendix.
>
> -  _On line 468, you introduce the symbol c_i but it is not clear where is comes from. Is it ith dimension of CLS vector?_
>
> Thanks for pointing this out. Yes, $c$ refers to the `CLS` vector, and $c_i$ refers to the $i^{th}$ element of the `CLS` vector. We will be sure to clarify this.
>
> -  _When constructing G_I, do you only keep the attention weights between tokens that belong to different parts of the input to get the bipartite graph? Are the attention scores renormalized?_
>
> Yes, we keep the attention weights between the tokens that belong to different parts to create the bipartite graph. No, they are not re-normalized.
>
> -  _While not necessary, one additional experiment that may strengthen the paper is to train a model with provided explanations to see if it returns superior accuracy compared to current models. This training may take a simple form of an additional loss supervising the attention matrix, etc._
>
> Thank you for this valuable suggestion. We will certainly explore this in future work.
>
> As for the reproducibility of the paper, the complete implementation with all hyper-parameters will be released along with the proposed dataset.
>
> We appreciate your efforts in a careful reading of the paper as reflected in the clarification questions. We hope that our responses and suggested revisions have bolstered the __soundness__ of our study as we made an effort to provide "extra support or details" for the "minor points" suggested and reflected in your soundness score.

---

### Meta-Review · Area_Chair_D3fB · 2023-09-19

**Recommendation:** 4

**Metareview:**

This work contributes a corpus of span interaction annotations for SNLI and FEVER instances. A primary goal of this work is to validate if NLP models use similar reasoning processes to humans. Reviewers appreciated the newly-created corpus, as well as the analyses in the paper.

---

### Decision · Program_Chairs · 2023-10-07

**Decision:**

Accept-Main

**Comment:**

This work contributes a corpus of span interaction annotations for SNLI and FEVER instances. A primary goal of this work is to validate if NLP models use similar reasoning processes to humans. Reviewers appreciated the newly-created corpus, as well as the analyses in the paper.